# CROSS-DOMAIN ADAPTATION FOR FEW-SHOT 3D SHAPE GENERATION

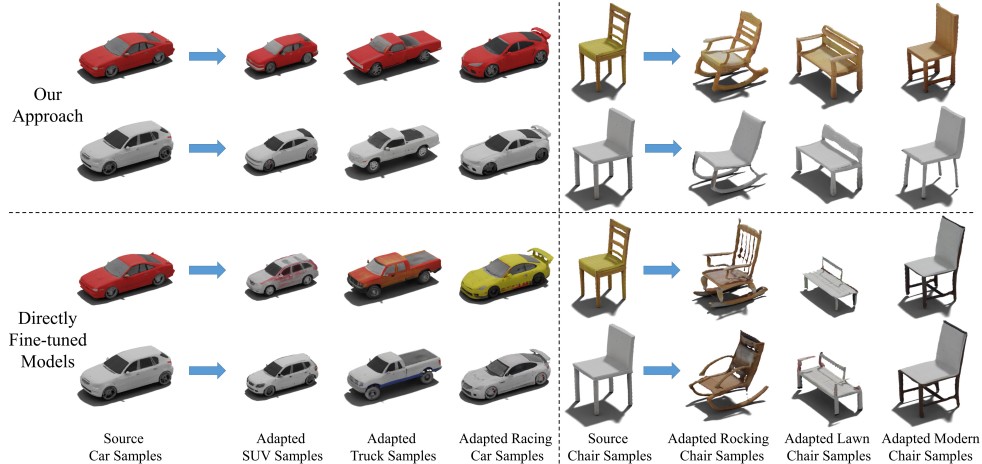

Figure 1: Given pre-trained 3D shape generative models, we propose to adapt them to target domains using a few target samples while preserving diverse geometry and texture information learned from source domains. Compared with directly fine-tuned models which tend to replicate the few-shot target samples instead of producing novel samples, our approach only needs the silhouettes of target samples as training data and achieves diverse generated shapes following target geometry distributions but different from target samples.

## ABSTRACT

Realistic and diverse 3D shape generation is helpful for a wide variety of applications such as virtual reality, gaming, and animation. Modern generative models learn from large-scale datasets and generate new samples following similar distributions. However, when training data is limited, deep neural generative networks overfit and tend to replicate training samples. Prior works focus on few-shot image generation to produce high-quality and diverse results using a few target images. Unfortunately, abundant 3D shape data is typically hard to obtain as well. In this work, we make the first attempt to realize few-shot 3D shape adaptation by adapting generative models pre-trained on large source domains to target domains. To relieve overfitting and keep considerable diversity, we propose to maintain the probability distributions of the pairwise relative distances between adapted samples at feature-level and shape-level during domain adaptation. Our approach only needs the silhouettes of few-shot target samples as training data to learn target geometry distributions and achieve generated shapes with diverse topology and textures. Moreover, we introduce several metrics to evaluate generation quality and diversity. The effectiveness of our approach is demonstrated qualitatively and quantitatively under a series of few-shot 3D shape adaptation setups.

## 1 INTRODUCTION

In recent years, 3D content has played significant roles in many applications, such as gaming, robotics, films, and animation. Currently, the most common method of creating 3D assets depends on manual efforts using specialized 3D modeling software like Blender and Maya, which is very time-consuming and cost-prohibitive to generate high-quality and diverse 3D shapes. As a result, the need for automatic 3D content generation becomes apparent.

During the past decade, image generation has been widely studied and achieved great success using generative models, including generative adversarial networks (GANs) (Goodfellow et al., 2014; Brock et al., 2019; Karras et al., 2019; 2020b; 2021), variational autoencoders (VAEs) (Kingma & Welling, 2013; Rezende et al., 2014; Vahdat & Kautz, 2020), autoregressive models (Van den Oord et al., 2016; Chen et al., 2018; Henighan et al., 2020), and diffusion models (Ho et al., 2020; Song & Ermon, 2020; Dhariwal & Nichol, 2021; Nichol & Dhariwal, 2021; Kingma et al., 2021). Compared with 2D images, 3D shapes are more complex and have different kinds of representations for geometry and textures. Inspired by the progress in 2D generative models, 3D generative models have become an active research area of computer vision and graphics and have achieved pleasing results in the generation of point clouds (Achlioptas et al., 2018; Yang et al., 2019; Zhou et al., 2021a), implicit fields (Chen & Zhang, 2019; Mescheder et al., 2019), textures (Pavllo et al., 2020; 2021; Richardson et al., 2023), and shapes (Gao et al., 2022; Liu et al., 2023). In addition, recent works based on neural volume rendering (Mildenhall et al., 2020) tackle 3D-aware novel view synthesis (Chan et al., 2021; 2022; Gu et al., 2022; Hao et al., 2021; Niemeyer & Geiger, 2021; Or-El et al., 2022; Schwarz et al., 2020; Xu et al., 2022; Zhou et al., 2021b; Schwarz et al., 2022).

Similar to 2D image generative models like GANs and diffusion models, modern 3D generative models require large-scale datasets to avoid overfitting and achieve diverse results. Unfortunately, it is not always possible to obtain abundant data under some circumstances. Few-shot generation aims to produce diverse and high-quality generated samples using limited data. Modern few-shot image generation approaches (Wang et al., 2018; Karras et al., 2020a; Mo et al., 2020; Wang et al., 2020; Li et al., 2020; Ojha et al., 2021; Zhao et al., 2022b; Zhu et al., 2022b;a; Zhao et al., 2023) adapt models pre-trained on large-scale source datasets to target domains using a few available training samples to relieve overfitting and produce adapted samples following target distributions. Nevertheless, few-shot 3D shape adaptation has yet to be studied, constrained by the complexity of 3D shape generation and the limited performance of early 3D shape generative models.

In this paper, we make the first attempt to study few-shot 3D shape adaptation pursuing high-quality and diverse generated shapes using limited data. We follow prior few-shot image generation approaches to adapt pre-trained source models to target domains using limited data. Since 3D shapes contain geometry and texture information, we need to clarify two questions: (i) what to learn from limited training data, and (ii) what to adapt from pre-trained source models to target domains. Naturally, we define two 3D shape domain adaptation setups: (i) geometry and texture adaptation (Setup A): the adapted models are trained to learn the geometry information of target data only and preserve the diversity of geometry and textures from source models, and (ii) geometry adaptation only (Setup B): the adapted models are trained to learn both the geometry and texture information of target data and preserve the diversity of geometry from source models only. Since the adaptation approach for Setup B is a degradation of the approach for Setup A, we mainly focus on Setup A in this paper and provide additional analysis and solution of setup B in Appendix C.

We design a few-shot 3D shape adaptation approach based on modern 3D shape GANs, which synthesize textured meshes with randomly sampled noises requiring 2D supervision only. Source models directly fine-tuned on limited target data cannot maintain generation diversity and produce results similar to training samples. As shown in Fig. 1, two different source samples become analogous after few-shot domain adaptation, losing diversity of geometry and textures. Therefore, we introduce a pairwise relative distances preservation approach to keep the probability distributions of geometry and texture pairwise similarities in generated shapes at both feature-level and shape-level during domain adaptation. In this way, the adapted models are guided to learn the common properties of limited training samples instead of replicating them. As a consequence, adapted models maintain similar generation diversity to source models and produce diverse results.

The main contributions of our work are concluded as follows:

- To our knowledge, we are the first to study few-shot 3D shape adaptation and achieve diverse generated shapes with arbitrary topology and textures.

- We propose a novel few-shot 3D shape adaptation approach to learn target geometry distributions using 2D silhouettes of extremely limited data (e.g., 10 shapes) while preserving diverse information of geometry and textures learned from source domains.

- We introduce several metrics to evaluate the quality and diversity of few-shot 3D shape generation and demonstrate the effectiveness of our approach qualitatively and quantitatively.

## 2   RELATED WORK

**3D Generative Models** Early works (Wu et al., 2016; Smith & Meger, 2017; Lunz et al., 2020; Gadelha et al., 2017; Henzler et al., 2019) extend 2D image generators to 3D voxel grids directly but fail to produce compelling results with high resolution due to the large computational complexity of 3D convolution networks. Other works explore the generation of alternative 3D shape representations, such as point clouds (Achlioptas et al., 2018; Yang et al., 2019; Zhou et al., 2021a) and implicit fields (Chen & Zhang, 2019; Mescheder et al., 2019). Following works generate meshes with arbitrary topology using autoregressive models (Nash et al., 2020) and GANs (Luo et al., 2021). Meshdiffusion (Liu et al., 2023) first applies diffusion models to generate 3D shapes unconditionally. These works produce arbitrary topology only and need post-processing steps to achieve textured meshes that are compatible with modern graphics engines. DIBR (Chen & Zhang, 2019) and Textured3DGAN (Pavllo et al., 2020; 2021) synthesize textured 3D meshes based on input templated meshes, resulting in limited topology. GET3D (Gao et al., 2022) first proposes a 3D generative model to achieve arbitrary and diverse 3D geometry structures and textures using 2D images for supervision. DreamFusion (Poole et al., 2022) depends on a pre-trained 2D text-to-image diffusion model to perform text-to-3D synthesis.

**3D Shape Translation** LOGAN (Yin et al., 2019) and UNIST (Chen et al., 2022) realize 3D shape translation based on VAEs trained on abundant data from two domains. Then translators are trained to transfer samples from one domain to the other based on the latent space provided by the VAEs. They tackle a different task from this work and aim to build a translation between two domains. Our approach aims to produce diverse results given few-shot data. Besides, LOGAN and UNIST are not qualified for few-shot data since both VAEs and translators need enough data to avoid overfitting.

**Few-shot Generation** Modern generative models need large amounts of data to achieve high-quality and diverse results. When training data is limited to a few samples, deep generative models tend to overfit and replicate them instead of generating novel results. Few-shot generation aims to solve the overfitting problem of generative models when training data is limited. Domain adaptation is a mainstream choice to realize few-shot generation. The key idea is to preserve the diverse information provided by source models while learning the common features of a few real target samples. In this way, a generative model for target domains is obtained to avoid overfitting or replicating training samples. The network structures of adapted models are consistent with source models in most cases.

**Few-shot Image Generation** Existing few-shot image generation methods aim to produce high-quality images with great diversity utilizing a few samples. Most modern approaches follow the TGAN (Wang et al., 2018) method to adapt generative models pre-trained on large source domains, including ImageNet (Deng et al., 2009), FFHQ (Karras et al., 2019), and LSUN (Yu et al., 2015) et al., to target domains with limited data. Following methods can be roughly divided into data augmentation approaches (Tran et al., 2021; Zhao et al., 2020a;b; Karras et al., 2020a), model regularization (Li et al., 2020; Ojha et al., 2021; Zhao et al., 2022b; Zhu et al., 2022b; Xiao et al., 2022), and trainable parameters fixing (Noguchi & Harada, 2019; Mo et al., 2020; Wang et al., 2020). CDC (Ojha et al., 2021) proposes a cross-domain consistency loss for generators and patch-level discrimination to build a correspondence between source and target domains. This work first explores few-shot 3D shape adaptation and shares similar ideas of preserving diverse information provided by source models, achieving diverse textured 3D shapes using limited data. We design losses sharing similar formats with CDC but apply to both feature-level and shape-level information and make them adaptive to 3D shapes with a series of modifications. The task and approach contribute to the novelty of this work.

## 3   METHOD

Given 3D generative models pre-trained on large source domains, our approach adapts them to target domains by learning the common geometry properties of limited training data while maintaining the generation diversity of geometry and textures. Directly fine-tuned models tend to replicate training samples instead of producing diverse results since the deep generative networks are vulnerable to overfitting, especially when training data is limited. To this end, we propose to keep the probability distributions of the pairwise relative distances between adapted samples similar to source samples.

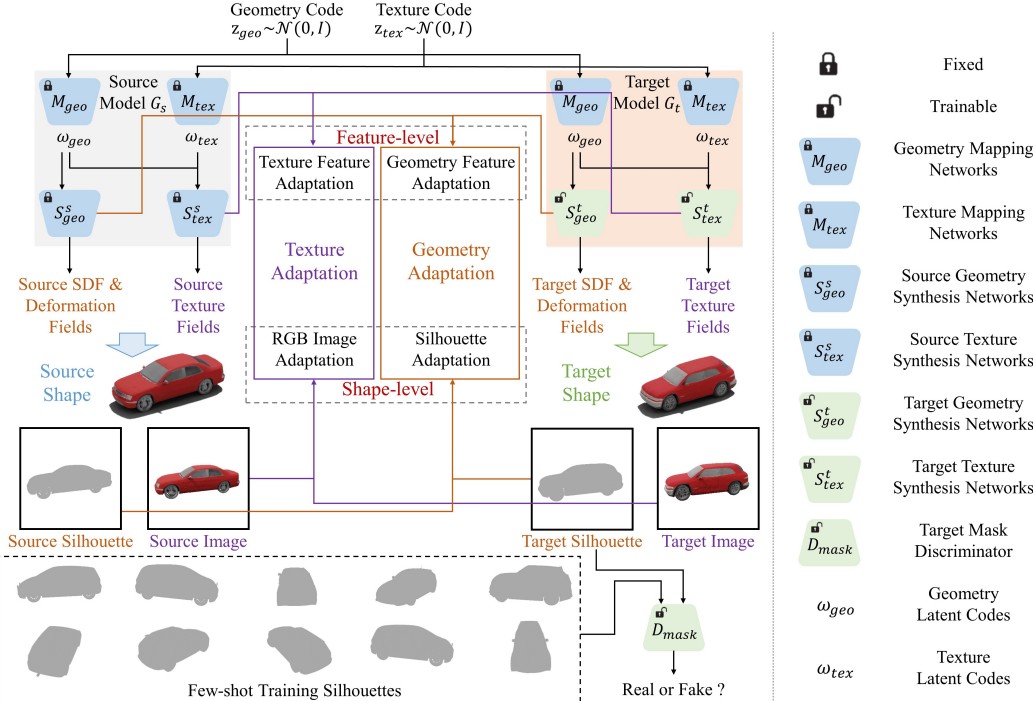

Figure 2: Overview of the proposed few-shot 3D shape generation approach using Cars → SUVs as an example: We maintain the distributions of pairwise relative distances between the geometry and textures of generated samples at feature-level and shape-level to keep diversity during domain adaptation. Only the silhouettes of few-shot target samples are needed as training data.

We employ the 3D shape generative model GET3D (Gao et al., 2022) to illustrate the proposed approach, as shown in Fig. 2. We first introduce GET3D briefly in Sec. 3.1. Our approach can be divided into geometry adaptation (Sec. 3.2) and texture adaptation (Sec. 3.3) using source models as reference. The silhouettes of target shapes are used as training data to learn geometry distributions.

## 3.1 PRELIMINARY: GET3D

GET3D is a 3D shape GAN trained on 2D images to generate 3D textured shapes. GET3D realizes arbitrary generation of topology and textures using the combination of geometry and texture generators. Both generators are composed of mapping networks $M$ and synthesis networks $S$. We empirically fix the mapping networks $M$ during domain adaptation in our approach. Ablations can be found in Appendix D. GET3D utilizes the differentiable surface representation DMTet (Shen et al., 2021) to describe geometry with signed distance fields (SDF) defined on deformation fields (Gao et al., 2020b;a). The texture generator uses mapped geometry and texture codes as inputs and generates texture fields for explicit meshes obtained by adopting DMTet for surface extraction. GET3D is trained with two 2D discriminators applied to RGB images and silhouettes, respectively.

GET3D is different from 3D-aware GANs and 3D diffusion models. Both 3D-aware GANs and GET3D need 2D images only as training data. 3D-aware GANs (Schwarz et al., 2020; Chan et al., 2021; 2022) generate novels views of 3D shapes but cannot extract 3D shapes directly. GET3D is the first randomly generative model trained on 2D images and synthesizing textured 3D shapes. Most 3D diffusion models (Liu et al., 2023; Nichol et al., 2022; Gupta et al., 2023; Chou et al., 2022; Shue et al., 2023) need 3D training data like meshes and point clouds since they need 3D ground truth to compute the reconstruction loss. Diffusion-based methods take up significantly larger computational costs, memory occupancy, and inference time. Our approach is implemented based on GET3D in this paper. However, it is not bound by certain network architectures of GET3D and can be applied to more powerful "real" 3D GANs in the future to achieve higher-quality results. As an analogy, early few-shot image generation works are implemented with BigGAN (Brock et al., 2019), but they can be applied to StyleGANs (Karras et al., 2019; 2020b) as well.

## 3.2 GEOMETRY ADAPTATION

We aim to guide adapted models to learn the common geometry properties of limited training samples while maintaining geometry diversity similar to source models. We propose to keep the probability distributions of pairwise relative distances between the geometry structures of adapted samples at feature-level and shape-level. We first sample a batch of geometry codes $\left\{z_{geo}^n\right\}_0^N$ following the standard normal distribution $\mathcal{N}(0, I)$ and get mapped geometry latent codes $\left\{\omega_{geo}^n\right\}_0^N$ using fixed geometry mapping networks $M_{geo}$. The probability distributions for the $i^{th}$ noise vector $z_{geo}^i$ in the source and target geometry generators at feature-level can be expressed as follows:

$$p_{geo,i}^{s,l} = sfm(\left\{sim(S_{geo}^{s,l}(\omega_{geo}^i), S_{geo}^{s,l}(\omega_{geo}^j))\right\}_{\forall i \neq j}), \tag{1}$$

$$p_{geo,i}^{t,l} = sfm(\left\{sim(S_{geo}^{t,l}(\omega_{geo}^i), S_{geo}^{t,l}(\omega_{geo}^j))\right\}_{\forall i \neq j}), \tag{2}$$

where $sfm$ and $sim$ represent the softmax function and cosine similarity between activations at the $l^{th}$ layer of the source and target geometry synthesis networks ($S_{geo}^s$ and $S_{geo}^t$) which generate SDF and deformation fields. Then we guide target geometry synthesis networks to keep similar probability distributions to source models during domain adaptation with the feature-level geometry loss:

$$\mathcal{L}_{geo}(S_{geo}^s, S_{geo}^t) = \mathbb{E}_{z_{geo}^i \sim \mathcal{N}(0,I)} \sum_{l,i} D_{KL}(p_{geo,i}^{t,l} || p_{geo,i}^{s,l}), \tag{3}$$

where $D_{KL}$ represents KL-divergence. Similarly, we use source and target silhouettes in place of the features in geometry synthesis networks to keep the pairwise relative distances of adapted samples at shape-level. For this purpose, we further sample a batch of texture codes $\left\{z_{tex}^n\right\}_0^N$ for shape generation. The probability distributions of shapes generated from the $i^{th}$ noise vectors ($z_{geo}^i$ and $z_{tex}^i$) by the source and target generators are given by:

$$p_{mask,i}^s = sfm(\left\{sim(Mask(G_s(z_{geo}^i, z_{tex}^i)), Mask(G_s(z_{geo}^j, z_{tex}^j)))\right\}_{\forall i \neq j}), \tag{4}$$

$$p_{mask,i}^t = sfm(\left\{sim(Mask(G_t(z_{geo}^i, z_{tex}^i)), Mask(G_t(z_{geo}^j, z_{tex}^j)))\right\}_{\forall i \neq j}), \tag{5}$$

where $G_s$ and $G_t$ are the source and target shape generators, $Mask$ represents the masks of 2D rendered shapes. We have the shape-level mask loss for geometry adaptation as follows:

$$\mathcal{L}_{mask}(G_s, G_t) = \mathbb{E}_{z_{geo}^i, z_{tex}^i \sim \mathcal{N}(0,I)} \sum_i D_{KL}(p_{mask,i}^t || p_{mask,i}^s). \tag{6}$$

## 3.3 TEXTURE ADAPTATION

In addition, we also encourage adapted models to preserve the texture information learned from source domains and generate target shapes with diverse textures. We still apply the pairwise relative distances preservation approach to relieve overfitting and keep the generation diversity of textures. Since the generated textures for explicit meshes contain both geometry and texture information, we propose to use textures in regions shared by two generated shapes to compute the pairwise relative distances of textures while alleviating the influence of geometry. In the same way, we use the randomly sampled geometry codes $\left\{z_{geo}^n\right\}_0^N$ and texture codes $\left\{z_{tex}^n\right\}_0^N$ and get mapped latent codes $\left\{\omega_{geo}^n\right\}_0^N$ and $\left\{\omega_{tex}^n\right\}_0^N$ with fixed geometry and texture mapping networks $M_{geo}$ and $M_{tex}$, respectively. The shared regions of two generated shapes produced by the source and adapted models are defined as the intersection of the masks of the 2D rendered shapes:

$$M_{i,j}^s = Mask(G_s(z_{geo}^i, z_{tex}^i)) \wedge Mask(G_s(z_{geo}^j, z_{tex}^j)) \ (i \neq j), \tag{7}$$

$$M_{i,j}^t = Mask(G_t(z_{geo}^i, z_{tex}^i)) \wedge Mask(G_t(z_{geo}^j, z_{tex}^j)) \ (i \neq j). \tag{8}$$

The probability distributions for the $i^{th}$ noise vectors ($z_{geo}^i$ and $z_{tex}^i$) in the source and target texture generators ($S_{tex}^s$ and $S_{tex}^t$) at feature-level can be expressed as follows:

$$p_{tex,i}^{s,m} = sfm(\left\{sim(S_{tex}^{s,m}(\omega_{geo}^i, \omega_{tex}^i) \otimes M_{i,j}^s, S_{tex}^{s,m}(\omega_{geo}^j, \omega_{tex}^j) \otimes M_{i,j}^s)\right\}_{\forall i \neq j}), \tag{9}$$

$$p_{tex,i}^{t,m} = sfm(\left\{sim(S_{tex}^{t,m}(\omega_{geo}^i, \omega_{tex}^i) \otimes M_{i,j}^t, S_{tex}^{t,m}(\omega_{geo}^j, \omega_{tex}^j) \otimes M_{i,j}^t)\right\}_{\forall i \neq j}), \tag{10}$$

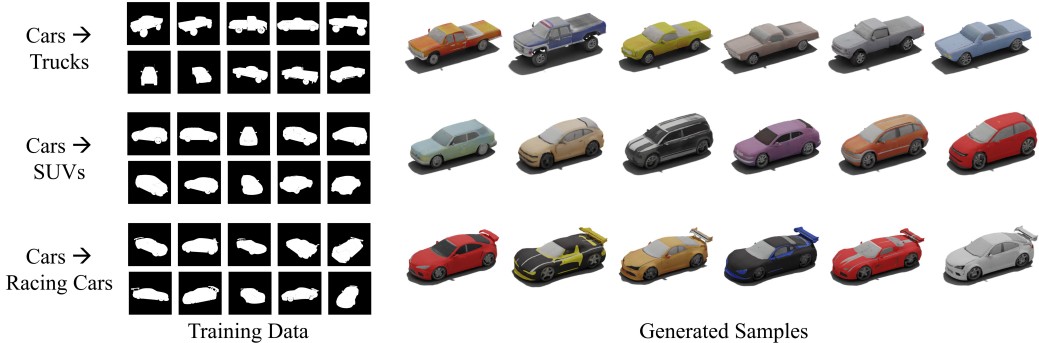

Figure 3: 10-shot generated shapes of our approach using ShapeNetCore Cars as the source domain.

where $\otimes$ and $sim$ represent the element-wise multiplication of tensors and cosine similarity between activations at the $m^{th}$ layer of the source and target texture synthesis networks. For shape-level texture adaptation, we use 2D rendered shapes of RGB formats in place of the features in texture synthesis networks to compute the probability distributions:

$$p_{rgb,i}^s = sfm(\left\{sim(RGB(G_s(z_{geo}^i, z_{tex}^i)) \otimes M_{i,j}^s, RGB(G_s(z_{geo}^j, z_{tex}^j)) \otimes M_{i,j}^s)\right\}_{\forall i \neq j}), \quad (11)$$

$$p_{rgb,i}^t = sfm(\left\{sim(RGB(G_t(z_{geo}^i, z_{tex}^i)) \otimes M_{i,j}^t, RGB(G_t(z_{geo}^j, z_{tex}^j)) \otimes M_{i,j}^t)\right\}_{\forall i \neq j}), \quad (12)$$

where $RGB$ represents the rendered RGB images of generated shapes. We have the feature-level texture loss and shape-level RGB loss for texture adaptation as follows:

$$\mathcal{L}_{tex}(S_{tex}^s, S_{tex}^t) = \mathbb{E}_{z_{geo}^i, z_{tex}^i \sim \mathcal{N}(0,I)} \sum_{m,i} D_{KL}(p_{tex,i}^{t,m} || p_{tex,i}^{s,m}), \quad (13)$$

$$\mathcal{L}_{rgb}(G_s, G_t) = \mathbb{E}_{z_{geo}^i, z_{tex}^i \sim \mathcal{N}(0,I)} \sum_{i} D_{KL}(p_{rgb,i}^t || p_{rgb,i}^s). \quad (14)$$

### 3.4 OVERALL OPTIMIZATION TARGET

Since adapted models are guided to learn the geometry information of training data, we only use the mask discriminator and apply the above-mentioned pairwise relative distances preservation methods to preserve diverse geometry and texture information learned from source domains. In this way, our approach only needs the silhouettes of few-shot target shapes as training data. The overall optimization target $\mathcal{L}$ of adapted models is defined as follows:

$$\mathcal{L} = \mathcal{L}(D_{mask}, G_t) + \mu\mathcal{L}_{reg} + \mu_1\mathcal{L}_{geo}(S_{geo}^s, S_{geo}^t) + \mu_2\mathcal{L}_{mask}(G_s, G_t) \\ + \mu_3\mathcal{L}_{tex}(S_{tex}^s, S_{tex}^t) + \mu_4\mathcal{L}_{rgb}(G_s, G_t). \quad (15)$$

Here $\mathcal{L}(D_{mask}, G_t)$ and $\mathcal{L}_{reg}$ represent the adversarial objective of silhouettes and regularization term of generated SDFs used in GET3D. More details of these two losses are added in Appendix B. $\mu, \mu_1, \mu_2, \mu_3, \mu_4$ are hyperparameters set manually to control the regularization levels.

## 4 EXPERIMENTS

We employ a series of few-shot 3D shape adaptation setups to demonstrate the effectiveness of our approach. We first show the qualitative results in Sec. 4.1. Then we introduce several metrics to evaluate quality and diversity quantitatively in Sec. 4.2. Finally, we ablate our approach in Sec. 4.3.

**Basic Setups** The hyperparameter of SDF regularization $\mu$ is set as 0.01 for all experiments. We empirically find $\mu_1 = 2e+4, \mu_2 = 5e+3, \mu_3 = 5e+3, \mu_4 = 1e+4$ to work well for the employed adaptation setups. We conduct experiments with batch size 4 on a single NVIDIA A40 GPU. The learning rates of the generator and discriminator are set as 0.0005. The adapted models are trained for 40K-60K iterations. The resolution of 2D rendered RGB images and silhouettes is $1024 \times 1024$. More details of implementation are added in Appendix J.

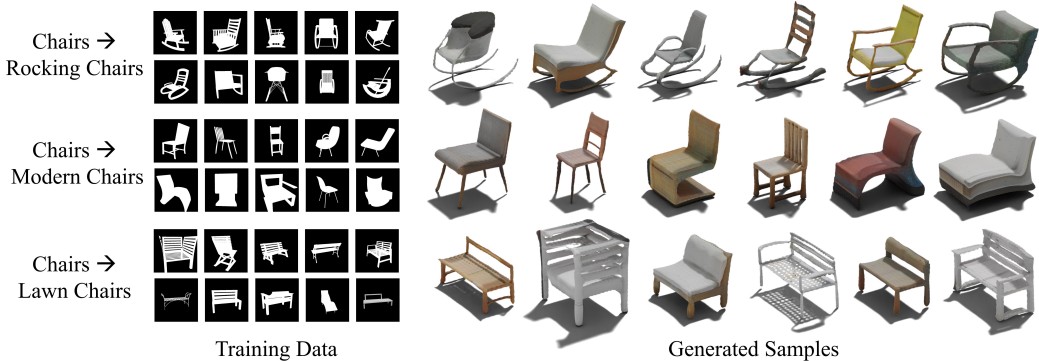

Figure 4: 10-shot generated shapes of our approach using ShapeNetCore Chairs as the source domain.

| Datasets | Approach | CD (↓) | Intra-CD (↑) | Pairwise-CD (↑) | Intra-LPIPS (↑) | Pairwise-LPIPS (↑) |
|---|---|---|---|---|---|---|
| Cars → SUVs | DFTM | 1.401 | $0.316 \pm 0.002$ | $0.513 \pm 0.001$ | $0.062 \pm 0.001$ | $0.063 \pm 0.012$ |
| | FreezeT | 1.553 | $0.240 \pm 0.005$ | $0.326 \pm 0.002$ | $0.055 \pm 0.002$ | $0.060 \pm 0.014$ |
| | Ours | **1.323** | $\mathbf{0.511 \pm 0.006}$ | $\mathbf{0.814 \pm 0.007}$ | $\mathbf{0.109 \pm 0.026}$ | $\mathbf{0.095 \pm 0.022}$ |
| Cars → Trucks | DFTM | 4.014 | $0.441 \pm 0.003$ | $0.689 \pm 0.003$ | $0.112 \pm 0.002$ | $0.119 \pm 0.024$ |
| | FreezeT | 4.175 | $0.412 \pm 0.006$ | $0.766 \pm 0.002$ | $0.120 \pm 0.003$ | $0.128 \pm 0.027$ |
| | Ours | **3.940** | $\mathbf{1.061 \pm 0.014}$ | $\mathbf{1.175 \pm 0.004}$ | $\mathbf{0.145 \pm 0.022}$ | $\mathbf{0.146 \pm 0.033}$ |
| Chairs → Lawn Chairs | DFTM | 40.559 | $4.001 \pm 0.005$ | $13.598 \pm 0.013$ | $0.165 \pm 0.029$ | $0.141 \pm 0.047$ |
| | FreezeT | 39.422 | $4.671 \pm 0.022$ | $19.269 \pm 0.024$ | $0.120 \pm 0.032$ | $0.165 \pm 0.040$ |
| | Ours | **38.661** | $\mathbf{5.852 \pm 0.031}$ | $\mathbf{22.989 \pm 0.022}$ | $\mathbf{0.278 \pm 0.040}$ | $\mathbf{0.166 \pm 0.054}$ |
| Chairs → Rocking Chairs | DFTM | 18.996 | $7.405 \pm 0.022$ | $15.312 \pm 0.011$ | $0.202 \pm 0.039$ | $0.203 \pm 0.037$ |
| | FreezeT | 18.503 | $5.541 \pm 0.014$ | $11.977 \pm 0.009$ | $0.203 \pm 0.046$ | $0.204 \pm 0.036$ |
| | Ours | **17.598** | $\mathbf{8.773 \pm 0.029}$ | $\mathbf{16.165 \pm 0.015}$ | $\mathbf{0.289 \pm 0.062}$ | $\mathbf{0.222 \pm 0.063}$ |

Table 1: Quantitative evaluation of our approach. Generated shapes of different approaches are synthesized from fixed noise inputs for fair comparison. CD scores are multiplied by $10^3$. The best results are highlighted in bold. Our approach performs better on both generation quality and diversity.

**Datasets** We use ShapeNetCore Cars and Chairs (Chang et al., 2015) as source datasets and sample several 10-shot shapes as target datasets, including (i) Trucks, (ii) Racing Cars, (iii) Sport Utility Vehicles (SUVs), corresponding to Cars and (iv) Rocking Chairs, (v) Modern Chairs, (vi) Lawn Chairs corresponding to Chairs. As shown in Fig. 3 and 4, the few-shot 3D shapes are processed to silhouettes using 24 randomly sampled and evenly distributed camera poses as training data. Ablations of the number of training samples and camera poses are provided in Appendix D.

**Baselines** Since few existing works explore few-shot 3D shape generation, we compare the proposed approach with directly fine-tuned models (DFTM) and fine-tuned models using fixed texture generators (FreezeT), including fixed texture mapping and texture synthesis networks.

## 4.1 QUALITATIVE EVALUATION

We visualize samples produced by our approach using source models pre-trained on ShapeNetCore Cars and Chairs in Fig. 3 and 4, respectively. Our approach only needs the silhouettes of few-shot training samples as target datasets to adapt source models to target domains while maintaining generation diversity of geometry and textures. In addition, we compare our approach with baselines using fixed noise inputs for intuitive comparison in Fig. 5. DFTM models replicate training samples and fail to keep generation diversity. FreezeT also fails to produce diverse textures since the mapped geometry codes influence the fixed texture synthesis networks. As a result, FreezeT models produce textured meshes similar to training samples under the guidance of RGB discriminators. Therefore, we further train FreezeT models without RGB discriminators or using source RGB discriminators. However, these two approaches still fail to preserve the diverse geometry and texture information of source models and cannot produce reasonable shapes. Our approach maintains the pairwise relative distances between generated shapes at feature-level and shape-level. It achieves high-quality and diverse adapted samples sharing geometry and texture information with source samples.

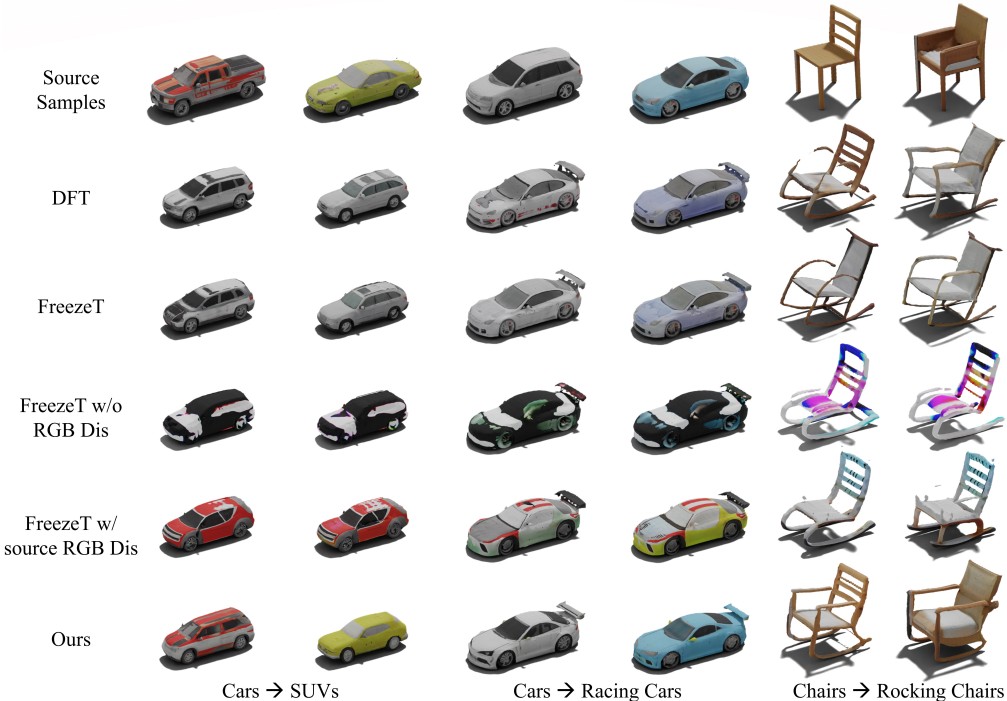

Figure 5: Visualized samples comparison on 10-shot Cars → SUVs, Cars → Racing Cars, and Chairs → Rocking Chairs. Results of different approaches are synthesized with fixed noise inputs.

## 4.2 QUANTITATIVE EVALUATION

**Evaluation Metrics** The generation quality of adapted models represents their capability to learn target geometry distributions. Chamfer distance (CD) (Chen et al., 2003) is employed to compute the distances of geometry distributions between 5000 adapted samples and target datasets containing relatively abundant target data to obtain reliable results. Besides, we design several metrics based on CD and LPIPS (Zhang et al., 2018) to evaluate the diversity of geometry and textures in adapted samples, which are computed in two ways: (i) pairwise-distances: we randomly generate 1000 shapes and compute the pairwise distances averaged over them, (ii) intra-distances (Ojha et al., 2021): we assign the generated shapes to one of the training samples with the lowest LPIPS distance and then compute the average pairwise distances within each cluster averaged over all the clusters. LPIPS results are averaged over 8 evenly distributed views of rendered shapes. Adapted models tending to replicate training samples may achieve fine pairwise-distances but only get intra-distances close to 0. Adapted models with great generation diversity achieve large values of both metrics.

The quantitative results of our approach are compared with baselines under several few-shot adaptation setups, as listed in Table 1. Our approach learns target geometry distributions better in terms of CD. Moreover, our approach also performs better on all the benchmarks of diversity, indicating its strong capability to produce diverse shapes with different geometry structures and textures.

## 4.3 ABLATION ANALYSIS

We provide ablation analysis to show the roles played by each component of our approach. In Fig. 6, we show the qualitative ablation analysis using 10-shot Chairs → Rocking Chairs as an example. Our full approach adapts source samples to target domains while preserving diverse geometry and texture information. Adapted models only using GAN loss with mask discrimination fail to maintain geometry diversity or produce high-quality shapes. Adding fixed source RGB discriminators results in texture degradation. Absence of the feature-level texture loss makes it harder for adapted models to preserve the texture information learned from source domains. Absence of shape-level RGB loss leads to repetitive textures and discontinuous shapes. As for the feature-level geometry and shape-level mask losses, their absence results in adapted samples sharing similar geometry structures and

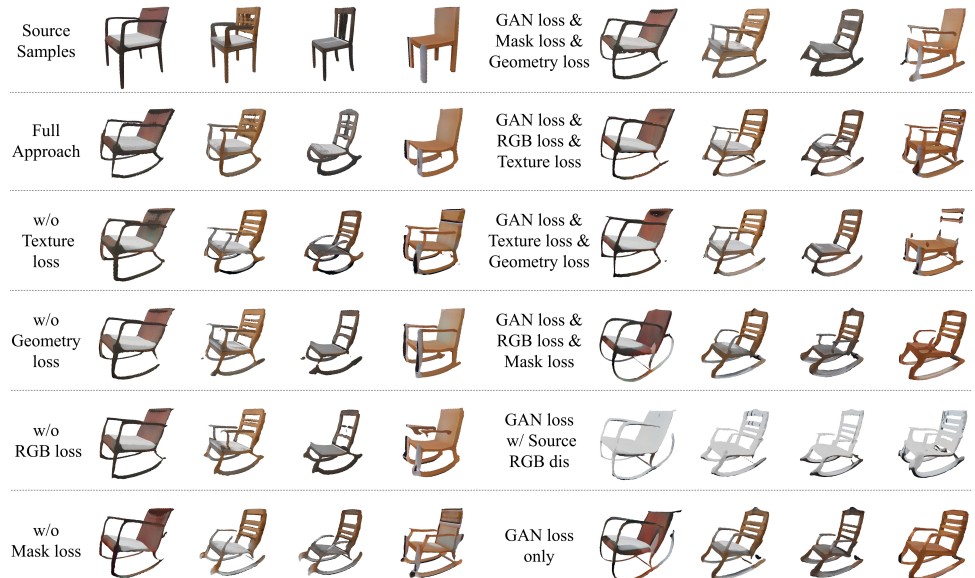

Figure 6: Qualitative ablations of our approach using 10-shot Chairs → Rocking Chairs as an example. Results of different approaches are synthesized with fixed noise inputs for intuitive comparison.

| Approach | CD (↓) | Intra-CD (↑) | Pairwise-CD (↑) | Intra-LPIPS (↑) | Pairwise-LPIPS (↑) |
|---|---|---|---|---|---|
| w/o Texture loss | 18.178 | $8.054 \pm 0.028$ | $13.533 \pm 0.010$ | $0.221 \pm 0.013$ | $0.210 \pm 0.045$ |
| w/o Geometry loss | 18.409 | $7.551 \pm 0.019$ | $12.549 \pm 0.009$ | $0.271 \pm 0.023$ | $0.217 \pm 0.057$ |
| w/o RGB loss | 17.762 | $7.207 \pm 0.018$ | $13.124 \pm 0.010$ | $0.211 \pm 0.006$ | $0.213 \pm 0.034$ |
| w/o Mask loss | 18.275 | $6.878 \pm 0.014$ | $12.435 \pm 0.008$ | $0.248 \pm 0.010$ | $0.208 \pm 0.010$ |
| Full Approach | **17.598** | $\mathbf{8.773 \pm 0.029}$ | $\mathbf{16.165 \pm 0.015}$ | $\mathbf{0.289 \pm 0.062}$ | $\mathbf{0.222 \pm 0.063}$ |

Table 2: Quantitative ablations of the proposed approach using 10-shot Chairs → Rocking Chairs as an example. The full approach performs better on both generation quality and diversity.

incomplete shapes. We also add ablations using geometry and mask losses, texture and RGB losses, feature-level losses, and shape-level losses, respectively. None of these approaches get compelling results. Incomplete geometry structures and low-quality textures can be found in their adapted samples. As shown in Table 2, the full approach achieves the best quantitative results on both generation quality and diversity. Without feature-level geometry loss or shape-level mask loss, adapted models perform worse on geometry diversity in terms of Intra-CD and Pairwise-CD. Similarly, adapted models perform worse on texture diversity in terms of Intra-LPIPS and Pairwise-LPIPS without feature-level texture loss or shape-level RGB loss. More ablations are added in Appendix D.

## 5 CONCLUSION AND LIMITATIONS

This paper first explores few-shot 3D shape adaptation. We introduce a novel domain adaptation approach to produce 3D shapes with diverse topology and textures. The relative distances between generated samples are maintained at both feature-level and shape-level. We only need the silhouettes of few-shot target samples as training data to learn target geometry distributions while keeping diversity. Our approach is implemented based on GET3D to demonstrate its effectiveness. However, it is not constrained by specific network architectures and can be combined with more powerful 3D shape generative models to produce higher-quality results in the future. Despite the compelling results of our approach, it still has some limitations. For example, it is mainly designed for related source/target domains. Extending our approach to unrelated domain adaptation would be promising. Nevertheless, we believe this work takes a further step towards democratizing 3D content creation by transferring knowledge in available source models to fit target distributions using few-shot data.

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

## SUPPLEMENTARY MATERIAL

We provide a detailed supplementary to help readers further understand our work and make this paper more convincing. The supplementary materials are organized as follows:

- Appendix A: **Broader Impact**

  Broader impact of this work.

- Appendix B: **More Details of GET3D**

  A more detailed introduction of GET3D, including the network architectures, training losses, and generated samples.

- Appendix C: **Geometry Adaptation Only**

  The discussion of the other few-shot 3D shape adaptation setup: geometry adaptation only (Setup B). Method and experiments (including qualitative and quantitative evaluation) are provided.

- Appendix D: **Supplementary Ablations**

  Supplementary qualitative and quantitative ablations of our approach are provided in this section.

- Appendix E: **Our Approach on Larger Domain Gaps**

  Experiments of our approach on larger domain gaps like Table → Modern Chairs.

- Appendix F: **Comparison with DreamBooth**

  Comparison between our approach on DreamBooth.

- Appendix G: **Additional Baseline Mine3D**

  We add an additional baseline Mine3D inspired by MineGAN (Wang et al., 2020). Qualitative and quantitative results are provided.

- Appendix I: **Novelty Analysis**

  Novelty analysis of our work.

- Appendix H: **More Details of Datasets**

  Detailed introduction of the datasets used in this paper.

- Appendix J: **More Details of Implementation**

  Details of the implementation of our approach, including method design, hyperparameters settings, and training details.

- Appendix K **Evaluation with CLIP**

  CLIP-based evaluation of the samples produced by our approach.

- Appendix L: **Supplementary Related Works**

  We add additional works related to this paper as reference.

- Appendix M: **Computational Cost**

  Computational cost statistics of baselines and our approach.

- Appendix N: **More Visualized Results**

  We provide abundant visualized results of our approach to make this paper more convincing.

**Reproducibility:** See the code provided in the submitted compressed file.

## A  BROADER IMPACT

We propose a novel approach for few-shot 3D shape generation, achieving high-quality and diverse 3D shape generation results using limited training data. Our approach is more prone to biases introduced by training data than typical artificial intelligence generative models since it only needs silhouettes of few-shot samples to train adapted models. The proposed approach is applicable to 3D shape generative models and not tailored for sensitive applications like generating human bodies. Therefore, we recommend practitioners to apply abundant caution when dealing with such applications to avoid problems of races, skin tones, or gender identities.

## B  MORE DETAILS OF GET3D

GET3D (Gao et al., 2022) is the first 3D shape generative model to produce textured meshes with arbitrary topology and textures. Here we add more details of the GET3D model. The mapping networks of GET3D are composed of 3D convolutional and fully connected networks. The synthesis networks for SDF and deformation fields are MLPs. As for the texture synthesis networks, GET3D uses generator network structures similar to StyleGAN2 to generate textures using triplane feature maps as inputs. GET3D also follows StyleGAN2 to use the same 2D discriminators and non-saturating GAN objective. Two 2D image discriminators are applied to RGB images and silhouettes, respectively. Given x representing an RGB image or a silhouette, the adversarial objective is defined as:

$$\mathcal{L}(D_x, G_t) = \mathbb{E}_{z \in \mathcal{N}}[g(D_x(R(G_t(z))))] + \mathbb{E}_{I_x \in p_x}\left[g(-D_x(I_x)) + \lambda||\nabla D_x(I_x)||_2^2\right], \quad (16)$$

where $g(u) = -log(1 + exp(-u))$, $p_x$ and $R$ represent the real image distributions and rendering functions for RGB images or silhouettes. In Eq. 15, we employ the discriminator for silhouettes as $\mathcal{L}(D_{mask}, G_t)$. The discriminator for RGB images used in GET3D is expressed as $\mathcal{L}(D_{rgb}, G_t)$. The regularization loss $\mathcal{L}_{reg}$ in Eq. 15 is designed to remove internal floating surfaces since GET3D aims to generate textured meshes without internal structures. $\mathcal{L}_{reg}$ is defined as a cross-entropy loss between the SDF values of neighboring vertices (Munkberg et al., 2022):

$$\mathcal{L}_{reg} = \sum_{i,j \in \mathbb{S}_e, i \neq j} H(\sigma(s_i), sign(s_j)) + H(\sigma(s_j), sign(s_i)). \quad (17)$$

Here $H$ and $\sigma$ represent binary cross-entropy loss and sigmoid function. $s_i, s_j$ are SDF values of neighboring vertices in the set of unique edges $\mathbb{S}_e$ in the tetrahedral grid. The regularization loss $\mathcal{L}_{reg}$ is applied to all the experiments (including ablation analysis) in this paper.

GET3D needs multi-view rendered RGB images and silhouettes with corresponding camera distribution parameters as training data. Therefore, it is evaluated with synthetic datasets such as ShapeNetCore (Chang et al., 2015) and TurboSquid (Turbosquid, Accessed: 2022-05-19). Future work may extend GET3D to single-view real-world datasets. If so, our approach can be applied to the advanced models to realize the few-shot generation of real-world 3D shapes using single-view silhouettes. Ablations of views used in domain adaptation is provided in Appendix D.

In Fig. 7, we provide generated samples of the officially released GET3D models trained on ShapNetCore Cars, Chairs, and Tables datasets. These models are used as source models in our experiments. GET3D generates shapes with arbitrary topology and textures. However, improvement room still exists for better results, such as incomplete textures of tires and failure of detailed structures generation in chairs. As a result, our approach produces some samples with incomplete textures of tires (as shown in Fig. 3) and cannot synthesize some detailed structures similar to some training samples. Our approach can be combined with better generative models in the future to achieve better visual effects.

## C  GEOMETRY ADAPTATION ONLY

In this section, we add the discussion of geometry adaptation only (Setup B) for target domains sharing similar textures like ambulances and police cars. Source models are trained to learn geometry and textures from limited training data under setup B. Adapted models only preserve the

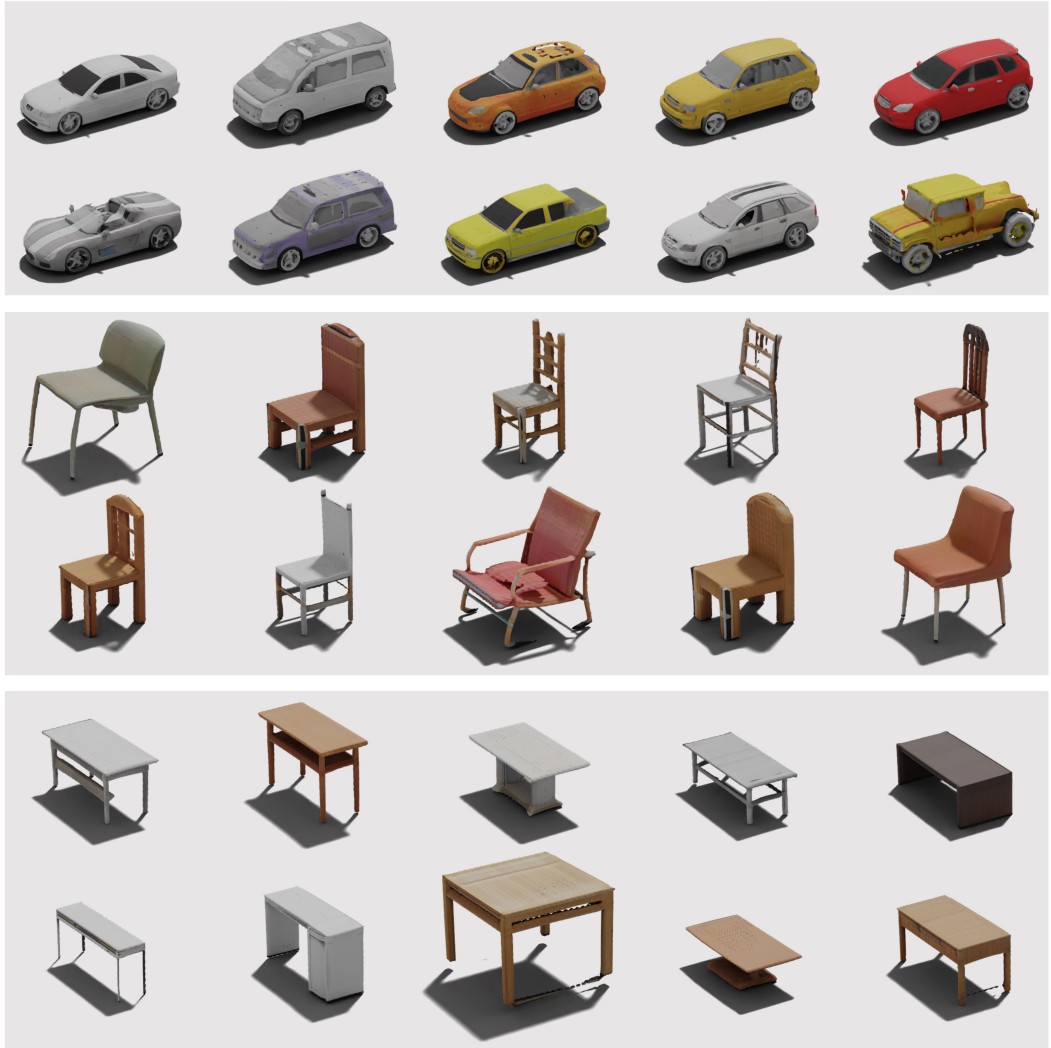

Figure 7: Generated shapes produced by the source GET3D (Gao et al., 2022) models trained on ShapeNetCore Cars and Chairs datasets (Chang et al., 2015)

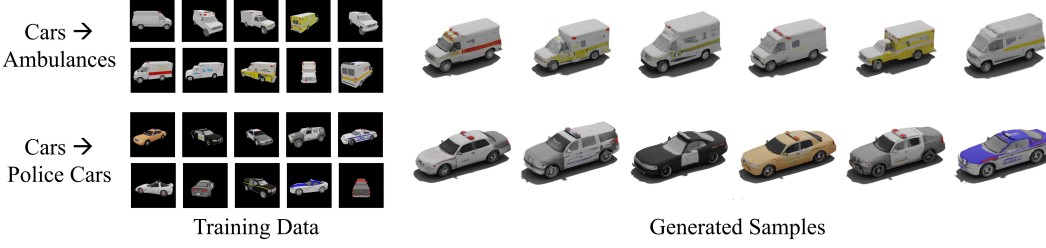

Cars → Ambulances

Cars → Police Cars

Training Data

Generated Samples

Figure 8: 10-shot generated shapes of our approach on Cars → Ambulances and Police Cars.

diversity of geometry learned from source domains. As for textures, we guide adapted models to fit the distributions of training samples.

**Method** The proposed adaptation approach under setup B has two differences compared with setup A (texture and geometry adaptation) discussed in our paper. Firstly, the feature-level texture loss and shape-level RGB loss are no longer needed. Secondly, generators are guided by the RGB discriminator to learn target texture distributions. Therefore, we need RGB images of rendered real samples

| Datastes | Approach | FID ($\downarrow$) | CD ($\downarrow$) |
|---|---|---|---|
| Cars $\rightarrow$ Ambulances | DFTM | **101.583** | 6.896 |
| | Ours | 103.708 | **5.963** |
| Cars $\rightarrow$ Police Cars | DFTM | 86.833 | 6.440 |
| | Ours | **74.958** | **5.616** |

Table 3: Quantitative evaluation of our approach on generation quality of geometry and textures.

| Datastes | Approach | Intra-CD ($\uparrow$) | Pairwise-CD ($\uparrow$) | Inra-LPIPS ($\uparrow$) | Pairwise-LPIPS ($\uparrow$) |
|---|---|---|---|---|---|
| Cars $\rightarrow$ Ambulances | DFTM | $0.300 \pm 0.002$ | $\mathbf{1.027 \pm 0.007}$ | $0.079 \pm 0.009$ | $0.083 \pm 0.017$ |
| | Ours | $\mathbf{0.558 \pm 0.004}$ | $0.638 \pm 0.006$ | $\mathbf{0.093 \pm 0.018}$ | $\mathbf{0.086 \pm 0.016}$ |
| Cars $\rightarrow$ Police Cars | DFTM | $0.426 \pm 0.003$ | $\mathbf{0.926 \pm 0.008}$ | $0.109 \pm 0.002$ | $0.108 \pm 0.017$ |
| | Ours | $\mathbf{0.902 \pm 0.005}$ | $0.902 \pm 0.006$ | $\mathbf{0.115 \pm 0.009}$ | $\mathbf{0.120 \pm 0.020}$ |

Table 4: Quantitative evaluation of our approach on generation diversity of geometry and textures.

as inputs for the RGB discriminator. The overall optimization target of adapted models under setup B is defined as follows:

$$\mathcal{L} = \mathcal{L}(D_{mask}, G_t) + \mathcal{L}(D_{rgb}, G_t) + \mu\mathcal{L}_{reg} + \mu_1\mathcal{L}_{geo}(S_{geo}^s, S_{geo}^t) + \mu_2\mathcal{L}_{mask}(G_s, G_t) \quad (18)$$

We follow GET3D to set $\mu = 0.01$ and empirically find $\mu_1$ and $\mu_2$ ranging from 2e+3 to 1e+4 appropriate for the adaptation setups used in our paper.

**Datasets** We use ShapeNetCore Cars (Chang et al., 2015) as source datasets and sample two 10-shot shapes as target datasets, including Police Cars and Ambulances. The 3D shapes are rendered as silhouettes and RGB images using 24 randomly sampled and evenly distributed camera poses as training data.

**Basic Setups** The basic setups of experiments under setup B are consistent with those under setup A (see Sec. 4), including hyperparameters, batch size, learning rates, resolution, and training hardware.

**Evaluation** We provide qualitative and quantitative results of our approach to demonstrate its effectiveness under setup B. As shown in Fig. 8, our approach produces ambulances and police cars with diverse topology using few-shot training samples qualitatively. For quantitative evaluation, we further add FID (Heusel et al., 2017) to evaluate the generation quality. FID results are averaged over 24 views of rendered shapes. The quantitative results are listed in Tables 3 and 4. Compared with DFTM models, our approach performs better on learning target geometry distributions in terms of CD. As for FID, our approach achieves better results on Cars $\rightarrow$ Police Cars and gets results close to the DFTM model on Cars $\rightarrow$ Ambulances. Besides, our approach achieves greater generation diversity in terms of Intra-CD and Intra-LPIPS. DFTM models get better results on Pairwise-CD and results close to our approach on Pairwise-LPIPS but get apparently worse results on intra-distances, indicating that they are overfitting to few-shot training samples and tend to replicate them instead of producing diverse results. We do not include FreezeT models for comparison under setup B since the adapted models need to learn the texture information from limited training samples.

## D    SUPPLEMENTARY ABLATIONS

**Ablations of Fixed Mapping Networks** As illustrated in Sec. 3, the geometry and texture mapping networks $M_{geo}$ and $M_{tex}$ are fixed during domain adaptation. We propose this design to isolate the geometry and texture adaptation since the texture synthesis networks need the mapped geometry codes as inputs. Without fixed mapping networks, fine-tuned geometry mapping networks would influence the texture adaptation process. We add ablations of fixed mapping networks under different adaptation setups and provide qualitative samples in Fig. 9. The low-quality adapted samples show blurred textures and fail to preserve the diverse texture information of source samples.

**K-shot Ablations** We add the ablations of the number of training samples. We empirically find that directly fine-tuned models produce diverse samples with about 50 samples but get limited diversity with fewer. Therefore, we use 10 training samples in our paper. Here we add the experiments of 5-shot, 3-shot, and 1-shot adaptation and get quantitative results as follows: Since intra-metrics are computed based on training samples, we only provide pairwise-metrics of different settings of

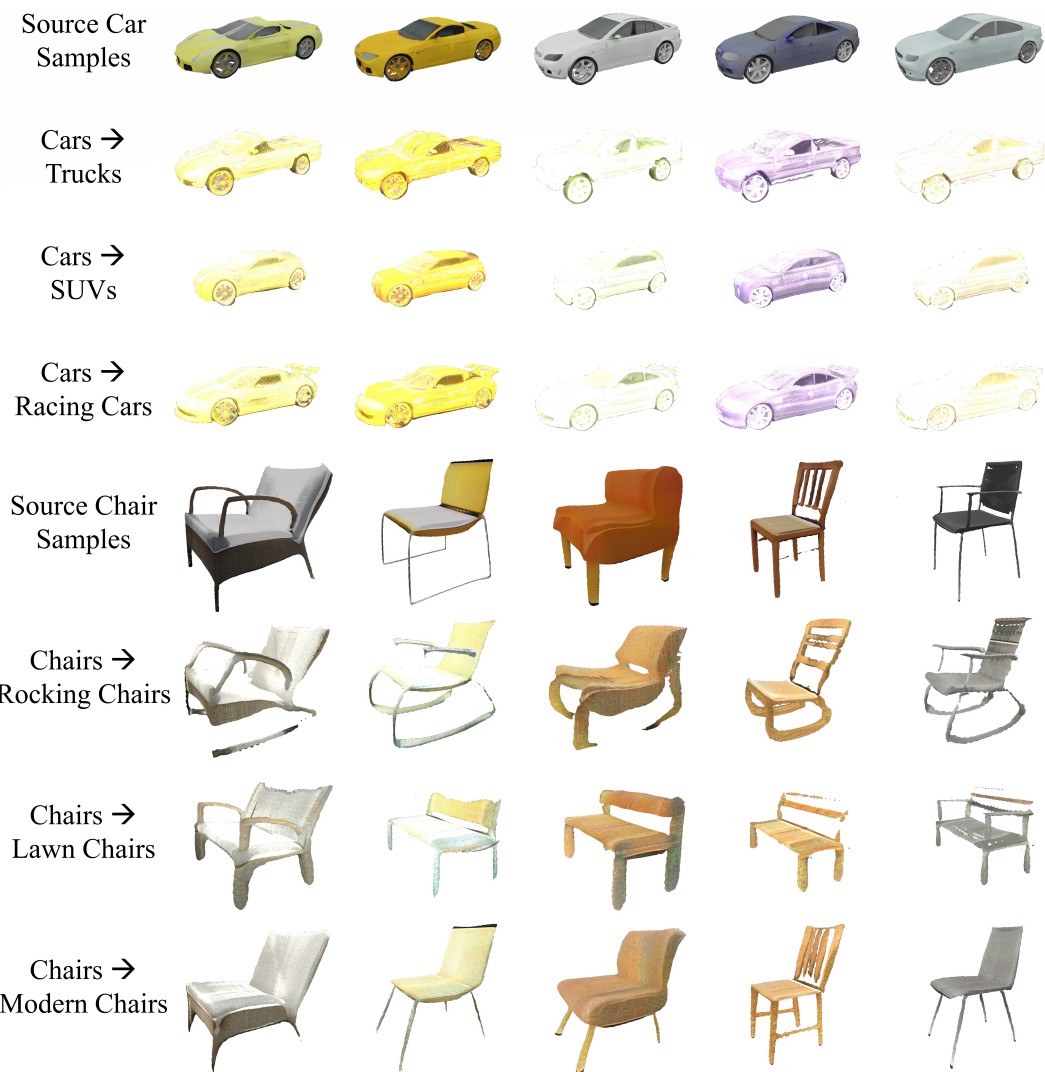

Figure 9: Qualitative ablations of fixed mapping networks during domain adaptation. Without fixed mapping networks, our approach fails to preserve the diverse texture information of source samples and produces blurred textures.

| Datasets | CD ($\downarrow$) | Pairwise-CD ($\uparrow$) | Pairwise-LPIPS ($\uparrow$) |
|---|---|---|---|
| 10-shot SUVs | **1.323** | **0.814 ± 0.007** | **0.095 ± 0.022** |
| 5-shot SUVs | 1.400 | 0.747 ± 0.008 | 0.094 ± 0.020 |
| 3-shot SUVs | 1.423 | 0.562 ± 0.015 | 0.090 ± 0.023 |
| 1-shot SUVs | 1.519 | 0.530 ± 0.012 | 0.086 ± 0.024 |

Table 5: Quantitative ablations of the number of training samples using few-shot Cars → SUVs as an example.

training samples in Table 5 for fair comparison. As the training samples decrease, the learning of target distributions and generation diversity of geometry structures become worse compared with 10-shot adaptation. The qualitative examples are shown in Fig. 10. Our approach still maintains a degree of diversity even with a single training sample, as shown in the qualitative results.

**K-view Ablations** We also explore the influence of the views of rendered samples on our approach. In this paper, we follow GET3D to use 24 randomly sampled views. Here we add 10-shot experiments using 18, 12, 6, and 1 randomly sampled views of each sample and get quantitative evaluation in Table 6.





Figure 10: Qualitative ablations of different numbers of training samples using Cars → SUVs as an example.

Figure 11: Qualitative ablations of different views of training samples using 10-shot Cars → SUVs as an example.

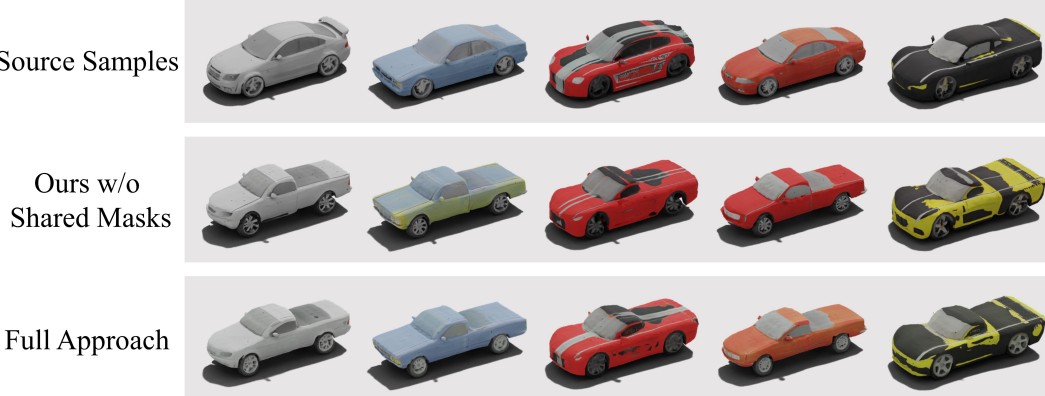

Figure 12: Qualitative ablations of shared masks applied to the feature-level texture loss and shape-level RGB loss using 10-shot Cars → Trucks as an example. The generated shapes of different approaches are synthesized with fixed noise inputs for intuitive comparison.

With fewer views, the learning of target distributions is biased as shown by the worse CD results. Besides, the diversity degrades inevitably as shown by the intra and pairwise metrics. The qualitative results are shown in Fig. 11. With fewer views of training samples (like 6-18 views), our approach generates plausible results using the prior knowledge of source models. For 1 view training, our approach gets some low-quality samples with unreasonable shapes (e.g., sharp car heads). However, our approach still shows a strong ability of maintaining quality and diversity using fewer views of training samples. Considering that GET3D needs more views (24 or 100 used in the GET3D (Gao et al., 2022) paper) to generate plausible results, our approach could serve as a strategy to improve generation quality and diversity by training from related source models using fewer views.

| Views | CD (↓) | Intra-CD (↑) | Pairwise-CD (↑) | Intra-LPIPS (↑) | Pairwise-LPIPS (↑) |
|---|---|---|---|---|---|
| 24 views | **1.323** | **0.511 ± 0.006** | **0.814 ± 0.007** | **0.109 ± 0.026** | **0.095 ± 0.022** |
| 18 views | 1.556 | 0.431 ± 0.010 | 0.796 ± 0.019 | 0.098 ± 0.017 | 0.088 ± 0.019 |
| 12 views | 1.623 | 0.415 ± 0.005 | 0.785 ± 0.002 | 0.090 ± 0.012 | 0.084 ± 0.015 |
| 6 views | 1.626 | 0.420 ± 0.005 | 0.789 ± 0.003 | 0.089 ± 0.028 | 0.083 ± 0.009 |
| 1 view | 1.755 | 0.427 ± 0.003 | 0.772 ± 0.014 | 0.083 ± 0.009 | 0.080 ± 0.008 |

Table 6: Quantitative ablations of the number of views using 10-shot Cars → SUVs as an example.

**Ablations of Shared Masks** In addition, we provide qualitative ablations for the shared masks used for feature-level texture loss and shape-level RGB loss computation in Fig. 12. Absence of shared masks causes geometry structures to bias the domain adaptation of textures, making the textures of adapted samples more different from source samples. Without shared masks, it's hard to preserve the diversity of textures influenced by geometry structures. As shown in Fig. 12, the absence of shared masks leads to more obvious changes of textures (e.g., colors and stripes) during adaptation compared with source samples. For example, the blue and orange source cars change into

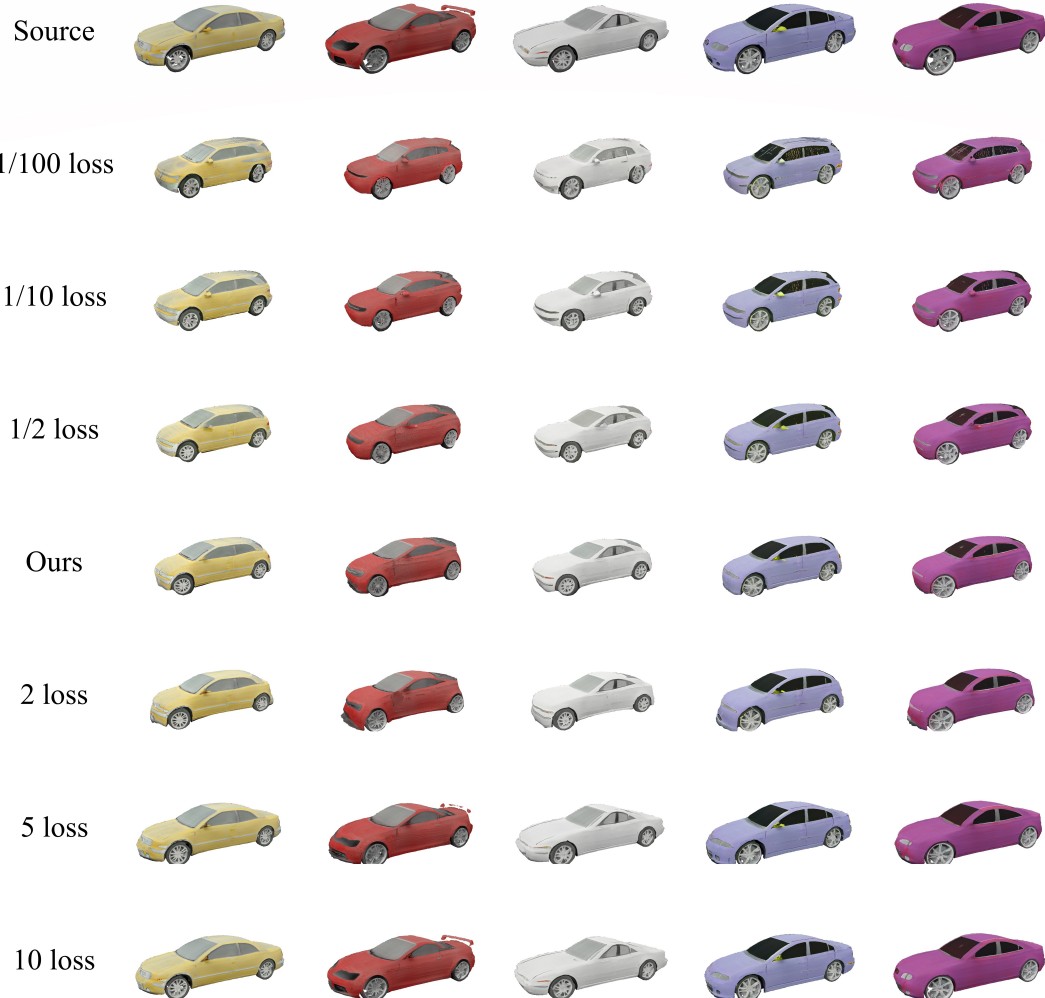

Figure 13: Qualitative ablations of the hyperparameters applied to the proposed adaptation losses using 10-shot Cars → SUVs as an example.

yellow-blue and red trucks during the 10-shot domain adaptation. The full approach applies shared masks to relieve the influence of geometry structures and achieves better preservation of the texture information in source models.

**Ablations of Hyperparameters** We add ablations of the hyperparameters applied to the proposed four adaptation losses. We use different values of hyperparameters and provide qualitative results using 10-shot Cars → SUVs in Fig. 13. Too large values of hyperparameters prevent adapted models from learning target distributions, resulting in results similar to source samples. Too small values of hyperparameters lead to diversity degradation of geometry and textures. We empirically recommend hyperparameters $\mu_1, \mu_2, \mu_3, \mu_4$ ranging from 2e+3 to 1e+4 for adaptation setups used in this paper.

# E  LARGER DOMAIN GAPS

We have conducted abundant experiments on related source/target domain adaptation like Cars → Trucks. In this section, we further add experiments on source/target domains with larger domain gaps. We employ two adaptation setups: Tables → Modern Chairs and Lawn Chairs trained on 10-shot silhouettes. As shown in Fig. 14, our approach is qualified for domain gaps like Tables → Chairs. Our approach adapts source table samples to chairs and retains considerable diversity. In addition, we add quantitative results to evaluate the generation diversity under different adaptation setups and report results in Table 7. Our approach achieves relatively lower diversity when adapting

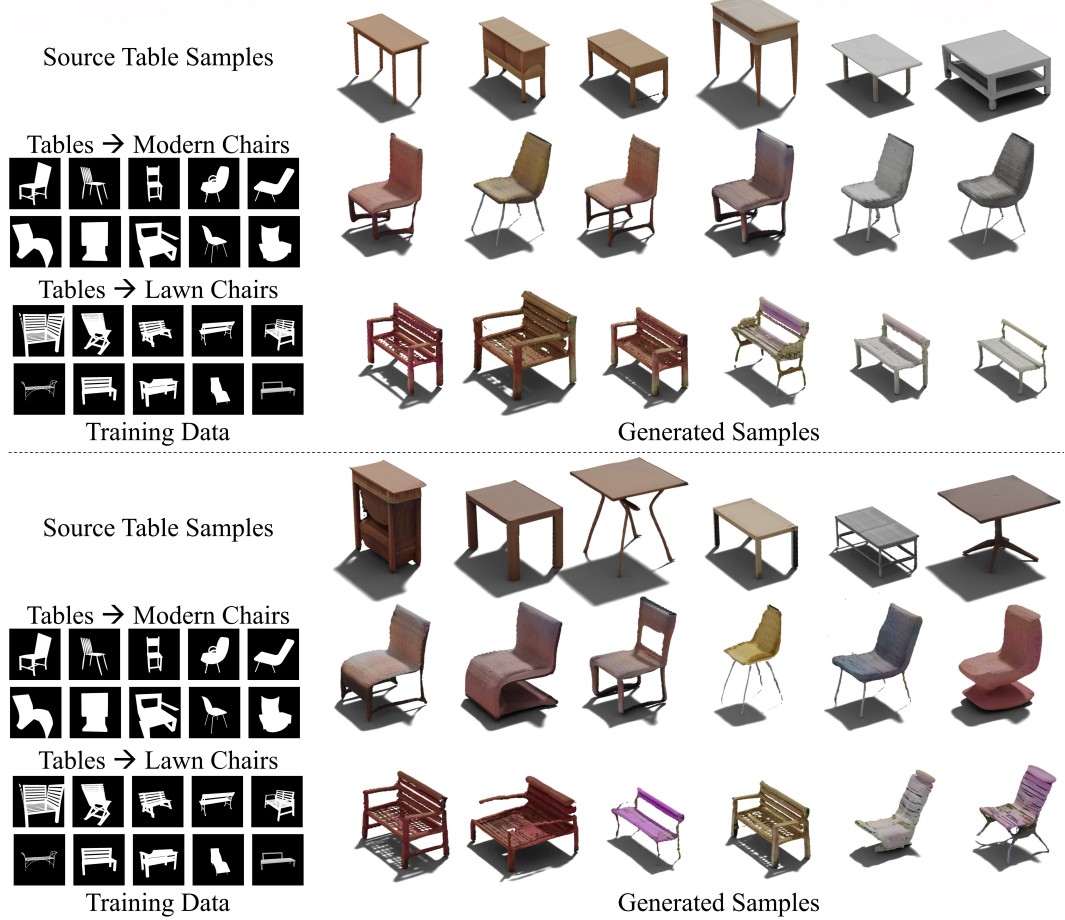

Figure 14: Visualized samples on 10-shot Tables → Modern Chairs and Lawn Chairs.

Tables to Modern and Lawn Chairs compared with adapting from Chairs. Despite that, our approach still achieves apparently greater generation diversity than directly fine-tuned models even with larger domain gaps, showing its ability to maintain diversity in few-shot 3D shape generation.

| Adaptation | Intra-CD (↑) | Intra-LPIPS (↑) |
|---|---|---|
| Chairs → Modern Chairs (directly fine-tuned) | $3.582 \pm 0.004$ | $0.149 \pm 0.023$ |
| Chairs → Modern Chairs (ours) | $\mathbf{5.011 \pm 0.022}$ | $\mathbf{0.254 \pm 0.045}$ |
| Tables → Modern Chairs (ours) | $4.735 \pm 0.021$ | $0.226 \pm 0.030$ |
| Chairs → Lawn Chairs (directly fine-tuned) | $4.001 \pm 0.005$ | $0.165 \pm 0.029$ |
| Chairs → Lawn Chairs (ours) | $\mathbf{5.852 \pm 0.031}$ | $\mathbf{0.278 \pm 0.040}$ |
| Tables → Lawn Chairs (ours) | $5.247 \pm 0.018$ | $0.242 \pm 0.036$ |

Table 7: Quantitative comparison between different adaptation setups. CD scores are multiplied by $10^3$.

## F    COMPARISON WITH DREAMBOOTH

DreamBooth (Ruiz et al., 2023) shares different targets with this work. DreamBooth is subject-driven and aims to synthesize novel scenes of the same subject in few-shot training samples, for example, novel views or novel contexts. The input includes the pre-trained Stable Diffusion Model and a reference set of the target subject. And the output is a fine-tuned Stable Diffusion Model, which could synthesize novel scenes of the target subject following corresponding text prompts. The key features of target subjects are preserved. For example, given a reference set of a dog, DreamBooth synthesizes images of the same dog.

Our work follows prior few-shot image generation methods (Wang et al., 2018; Mo et al., 2020; Wang et al., 2020; Li et al., 2020; Ojha et al., 2021; Zhao et al., 2022b) and is domain-driven. It is designed to generate high-quality and diverse samples of target domains. We aim to extract the common features of limited data and maintain generation diversity by adapting source samples to target domains. The subjects in adapted samples are a lot more diverse than few-shot training data. The input includes the source model pre-trained on a large source domain and few-shot samples of target domains. The output is a fine-tuned model which could generate target samples. For example, we use silhouettes of 10-shot SUVs as training data and adapt the source model of cars to the target domain of SUVs. The adapted model is not trained to synthesize SUVs in training data. Instead, it is trained to generate diverse SUVs (cars sharing common features with few-shot training data).

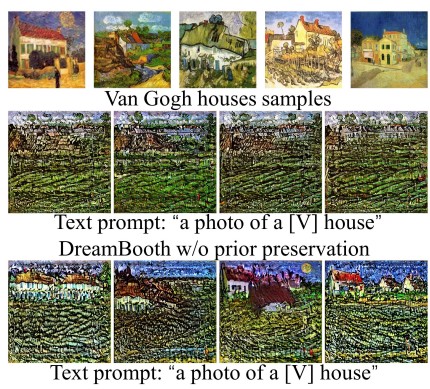

Van Gogh houses samples

Text prompt: "a photo of a [V] house"
DreamBooth w/o prior preservation

Text prompt: "a photo of a [V] house"
DreamBooth w/ prior "a photo of a house"

Figure 15: Stable Diffusion model fine-tuned by DreamBooth on 10-shot datasets.

Besides, 2D DreamBooth is not qualified for domain-driven tasks. For example, it cannot learn the domain knowledge (e.g., styles) from few-shot data. As shown in Fig. 15, we fine-tune the Stable Diffusion model on 10-shot Van Gogh houses dataset. We employ a unique identifier [V] to avoid using prior knowledge of target domains in the Stable Diffusion model. As shown in Fig. 15, DreamBooth without prior preservation severely overfits and produces low-quality samples. The full DreamBooth approach still overfits and gets limited quality and diversity. They tend to preserve the subjects in training samples instead of generating diverse samples following similar distributions. Therefore, applying DreamBooth to 3DGen (Gupta et al., 2023), which employs a diffusion model to encode input images as features for a triplane VAE to produce textured 3D shapes based on point cloud inputs, may fail to accomplish the few-shot 3D shape adaptation task tackled by our approach.

## G  ADDITIONAL BASELINE MINE3D

Similar to MineGAN (Wang et al., 2020), we add another baseline of "Mine3D" by adding two additional 4-layer MLPs using the texture and geometry latent codes as inputs, respectively. The whole generator is fixed. However, similar to MineGAN, it fails to build cross-domain correspondence like our approach and still overfits and replicates training samples. Qualitative examples are shown in Fig. 16. We use the same 10-shot adaptation setting as our approach. We choose two samples from training data to show that Mine3D produces samples very similar to training samples. The quantitative results are shown in Table 8. Mine3D gets worse quantitative results than our approach. We also tried to fix the RGB discriminator of Mine3D but got blurred textures like FreezeT w/o RGB discriminator shown in the row of Fig. 5.

| Datasets | Approach | CD ($\downarrow$) | Intra-CD ($\uparrow$) | Pairwise-CD ($\uparrow$) | Intra-LPIPS ($\uparrow$) | Pairwise-LPIPS ($\uparrow$) |
|---|---|---|---|---|---|---|
| Cars $\rightarrow$ SUVs | DFTM | 1.401 | $0.316 \pm 0.002$ | $0.513 \pm 0.001$ | $0.062 \pm 0.001$ | $0.063 \pm 0.012$ |
| | FreezeT | 1.553 | $0.240 \pm 0.005$ | $0.326 \pm 0.002$ | $0.055 \pm 0.002$ | $0.060 \pm 0.014$ |
| | Mine3D | 1.470 | $0.328 \pm 0.005$ | $0.622 \pm 0.001$ | $0.075 \pm 0.002$ | $0.071 \pm 0.020$ |
| | Ours | **1.323** | **$0.511 \pm 0.006$** | **$0.814 \pm 0.007$** | **$0.109 \pm 0.026$** | **$0.095 \pm 0.022$** |
| Cars $\rightarrow$ Trucks | DFTM | 4.014 | $0.441 \pm 0.003$ | $0.689 \pm 0.003$ | $0.112 \pm 0.002$ | $0.119 \pm 0.024$ |
| | FreezeT | 4.175 | $0.412 \pm 0.006$ | $0.766 \pm 0.002$ | $0.120 \pm 0.003$ | $0.128 \pm 0.027$ |
| | Mine3D | 4.199 | $0.508 \pm 0.010$ | $0.824 \pm 0.002$ | $0.126 \pm 0.014$ | $0.130 \pm 0.022$ |
| | Ours | **3.940** | **$1.061 \pm 0.014$** | **$1.175 \pm 0.004$** | **$0.145 \pm 0.022$** | **$0.146 \pm 0.033$** |

Table 8: Quantitative results of Mine3D compared with other baselines and our approach.

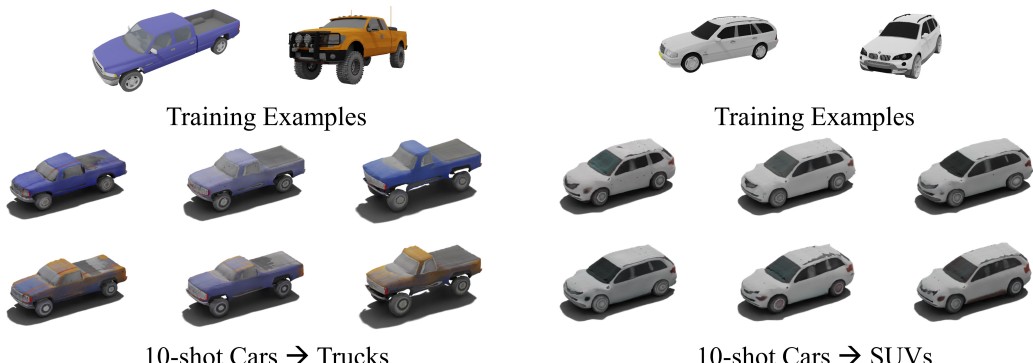

Figure 16: 10-shot generated shapes of Mine3D on 10-shot Cars → Trucks and SUVs.

## H  MORE DETAILS OF DATASETS

This paper employs several 10-shot datasets sampled from ShapeNetCore (Chang et al., 2015) as training data for few-shot 3D shape generation. As for the main experiments of our paper, we only need silhouettes of target samples as training data, as shown in Fig. 2. For the experiments of geometry adaptation only (Sec. C), rendered RGB images are also needed to train adapted models. The training datasets used in this paper are shown in Fig. 3, 4, and 8.

We employ CD (Chen et al., 2003) and FID (Heusel et al., 2017) as quantitative evaluation metrics for generation quality. Datasets containing relatively abundant data are applied for evaluation to obtain reliable results. The few-shot samples are excluded from the relatively abundant datasets to avoid the influence of overfitting. The relatively abundant Trucks, SUVs, Ambulances, Police Cars, Rocking Chairs, and Lawn Chairs datasets contain 40, 369, 73, 133, 87, and 78 samples.

The few-shot target shapes are not included in the training dataset of GET3D. The source GET3D models are pre-trained on the ShapeNetCore v1 dataset, while the target shapes are picked from the ShapeNetCore v2 dataset. We have checked that they are not included in the ShapeNetCore v1 dataset. Moreover, it doesn't influence our approach's evaluation even if the target shapes are directly picked from the source datasets. The source models still fail to produce diverse target samples, which is the target of the adapted models obtained through our approach. Instead, they generate lots of samples out of target domains. Taking trucks as the target domain, the source model is trained on cars in the ShapeNetCore v1 dataset. There are only 31 truck samples in 3533 car samples, making it hard to produce truck samples with the source model. Prior 2D GAN-based few-shot image generation works (Wang et al., 2020; 2018; Li et al., 2020; Zhu et al., 2022b;a; Ojha et al., 2021; Zhao et al., 2022b) directly use samples in FFHQ to adapt the source model pre-trained on FFHQ to a smaller target domain like babies and people wearing sunglasses.

## I  NOVELTY ANALYSIS

Our approach is composed of feature-level geometry loss, feature-level texture loss, shape-level mask loss, and shape-level RGB loss. Our approach is mainly inspired by contrastive learning methods (Oord et al., 2018; He et al., 2020; Chen et al., 2020). Similar approaches can be found in recent few-shot image generation approaches (Ojha et al., 2021; Zhu et al., 2022b;a) as well.

The proposed losses share similar formats with CDC (Ojha et al., 2021) and prior contrastive learning methods but still have differences. Firstly, CDC only uses features in generators for the computation of losses. Our approach uses both feature-level and shape-level (image-level) information and achieves higher-quality and more stable results compared with using feature-level losses only, as shown in the ablation results (Fig. 6). Secondly, our approach is designed for 3D shape generators and applies to geometry and texture information separately. Thirdly, we propose the concept of shared masks to alleviate the influence of geometry for texture adaptation, which helps our approach better preserve the texture information of source samples during domain adaptation. Besides, the novelty of this work also comes from the research topic. This is the first work to study the few-shot

3D shape adaptation task. In addition, we also introduce several metrics for few-shot 3D shape generation evaluation.

## J   MORE DETAILS OF IMPLEMENTATION

The proposed approach is implemented based on the official code of GET3D (Gao et al., 2022). The setups of adapted models are consistent with those of the officially released source models trained on ShapeNetCore Cars and Chairs (Chang et al., 2015). The geometry and texture synthesis networks are composed of 2-layers MLP networks. We concatenate the output features of the first layers in the synthesis networks of SDFs and deformation fields for feature-level geometry loss computation since the output features of the second layers have different sizes for SDFs and deformation fields. We also use the features in the synthesis networks of SDFs and deformation fields separately for feature-level geometry loss computation. Unfortunately, it is more time-consuming and fails to produce better results. For feature-level texture loss computation, we use the output features of the second layers in the texture synthesis network, which has the same resolution as the generated shapes. Therefore, we can directly apply the shared masks of generated shapes to the texture features.

The weights in target models are initialized to source models. We set the learning rates of the generator and discriminator as 0.0005, which is lower than the learning rates of source models (0.002), to realize more refined adaptation processes. We set the hyperparameters of the proposed losses ($\mu_1, \mu_2, \mu_3, \mu_4$) equally for adaptation from Cars and Chairs and achieve high-quality results. Different hyperparameters can be tried to obtain compelling results under other adaptation setups. We train adapted models with batch size 4 on a single NVIDIA A40 GPU (45GB GPU memory). Our approach needs about 20 GB GPU memory for the image resolution of $1024 \times 1024$. The standard deviations of pairwise-distance and intra-distance results listed in Tables 1, 4, and 2 are computed across shape pairs picked from generated samples and 10 clusters (the same number as few-shot training samples), respectively.

## K   EVALUATION WITH CLIP

We employ CLIP (Radford et al., 2021b) to evaluate the domain gap between the generated samples produced by our approach and target domains. We randomly synthesize 1024 3D shapes for every target domain and sample 24 2D RGB images with different angles for every sample. Then we use the CLIP image encoder to encode the 2D images into embeddings and the CLIP text encoder to encode the text prompt corresponding to the target domain (e.g., "a photo of a truck", "a photo of a rocking chair") into embeedings. We compute the cosine similarity between these two embeddings as the CLIP-based text-image similarity metric. The results of few-shot training data are provided as reference, and samples produced by the source models are used for comparison. As shown in Table 9, our approach achieves text-image similarity similar to few-shot datasets and outperforms source samples significantly. For ambulances and police cars, our approach guides adapted models to learn both geometry structures and textures, resulting in better text-image similarity compared with the other 4 adaptation setups of learning geometry structures only.

| Target Domains | Source Samples | Adapted Samples (ours) | Few-shot Data (Reference) |
|---|---|---|---|
| Trucks | 0.2041 | **0.2477** | 0.2568 |
| SUVs | 0.2183 | **0.2548** | 0.2686 |
| Rocking Chairs | 0.2549 | **0.2979** | 0.3120 |
| Lawn Chairs | 0.2280 | **0.2742** | 0.2891 |
| Ambulances | 0.2175 | **0.2913** | 0.2961 |
| Police Cars | 0.2323 | **0.2737** | 0.2730 |

Table 9: CLIP-based text-image similarity results of our approach on several datasets compared with source samples and few-shot training data.

## L   SUPPLEMENTARY RELATED WORKS

Here we provide supplementary works related to this paper as reference. MaskDis (Zhu et al., 2022b) proposes to regularize the discriminator using masked features. DDPM-PA (Zhu et al.,

2022a) first realizes few-shot image generation with diffusion models. Besides, other recent works have provided different research perspectives. RSSA (Xiao et al., 2022) proposes a relaxed spatial structural alignment method using compressed latent space derived from inverted GANs (Abdal et al., 2020). AdAM (Zhao et al., 2022a) and RICK (Zhao et al., 2023) achieve improvement in the adaptation of unrelated source/target domains. Research including MTG (Zhu et al., 2021), OSCLIP (Kwon & Ye, 2022), GDA (Zhang et al., 2022b), and DIFA (Zhang et al., 2022a) et al. explore single-shot GAN adaptation with the guidance of pre-trained CLIP (Radford et al., 2021a) image encoders.

HoloDiffusion (Karnewar et al., 2023b) proposes a 3D-aware generative diffusion model to reconstruct 3D-consistent objects using 2D posed images. HoloFusion (Karnewar et al., 2023a) generates photo-realistic 3D radiance fields by combining HoloDiffusion with a 2D super-resolution network.

## M  COMPUTATIONAL COST

Table 10 shows the computational cost of our approach under two adaptation setups using a single NVIDIA A40 GPU. We also ablate our approach to show the computational cost of each component. The adapted models are trained for about 40K-60K iterations in our experiments, costing about 4.4-6.5 and 3.8-5.7 hours under setup A (geometry and texture adaptation) and setup B (geometry adaptation only), respectively. DFTM under setup B is the same as training GET3D models directly. DFTM under setup A excludes the RGB discriminator. Compared with DFTM, the approach only using GAN loss includes the time cost by source models.

| Setups | Approaches | Time cost for 1K iterations |
|---|---|---|
| Setup A | DFTM | 228.83 |
| | GAN loss only | 272.27 |
| | GAN loss w/ Texture loss | 352.80 |
| | GAN loss w/ Geometry loss | 295.28 |
| | GAN loss w/ RGB loss | 291.62 |
| | GAN loss w/ Mask loss | 279.34 |
| | Full Approach | 392.67 |
| Setup B | DFTM | 281.15 |
| | GAN loss only | 316.55 |
| | GAN loss w/ Geometry loss | 344.82 |
| | GAN loss w/ Mask loss | 322.51 |
| | Full Approach | 340.38 |

Table 10: The time cost of our approach trained for 1K iterations in terms of seconds on a single NVIDIA A40 GPU (image resolution $1024 \times 1024$, batch size 4).

## N  MORE VISUALIZED RESULTS

In Fig. 17, we add generated shapes of different target domains rendered in multiple views. Our approach produces high-quality results different from the few-shot training samples. We employ the ShapeNetCore v1 (Chang et al., 2015) Tables datasets and sample two 10-shot target datasets: School Tables and Round Tables. The visualized results are shown in Fig. 18. Our approach also achieves high-quality and diverse samples under these adaptation setups. As supplements to generated samples shown in Fig. 3 and 4, we display more examples produced by our approach under several few-shot adaptation setups. Adapted samples obtained with the source models pre-trained on ShapeNetCore Cars and Chairs (Chang et al., 2015) are shown in Fig. 19 and 20, respectively.

We further add hundreds of randomly generated samples on 10-shot Cars $\rightarrow$ Racing Cars and SUVs in Fig. 21 to show that most of the samples produced by our approach are plausible. The quality and diversity of these samples make our work more convincing.

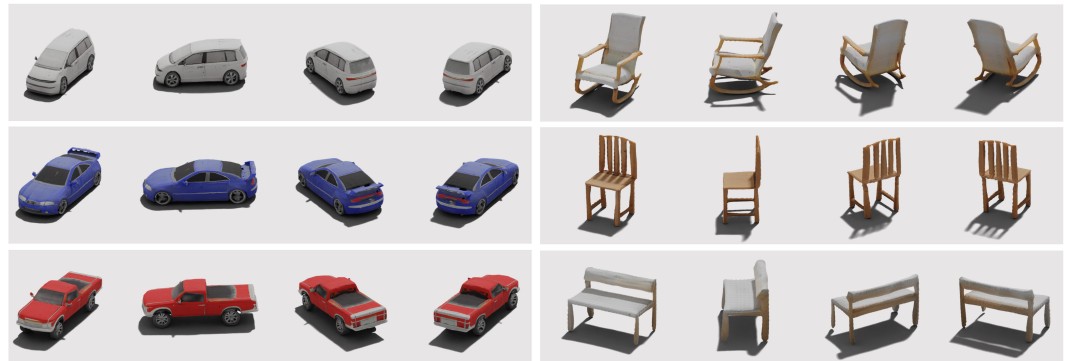

Figure 17: Multi-view rendered shapes produced by our approach on different 10-shot target domains.

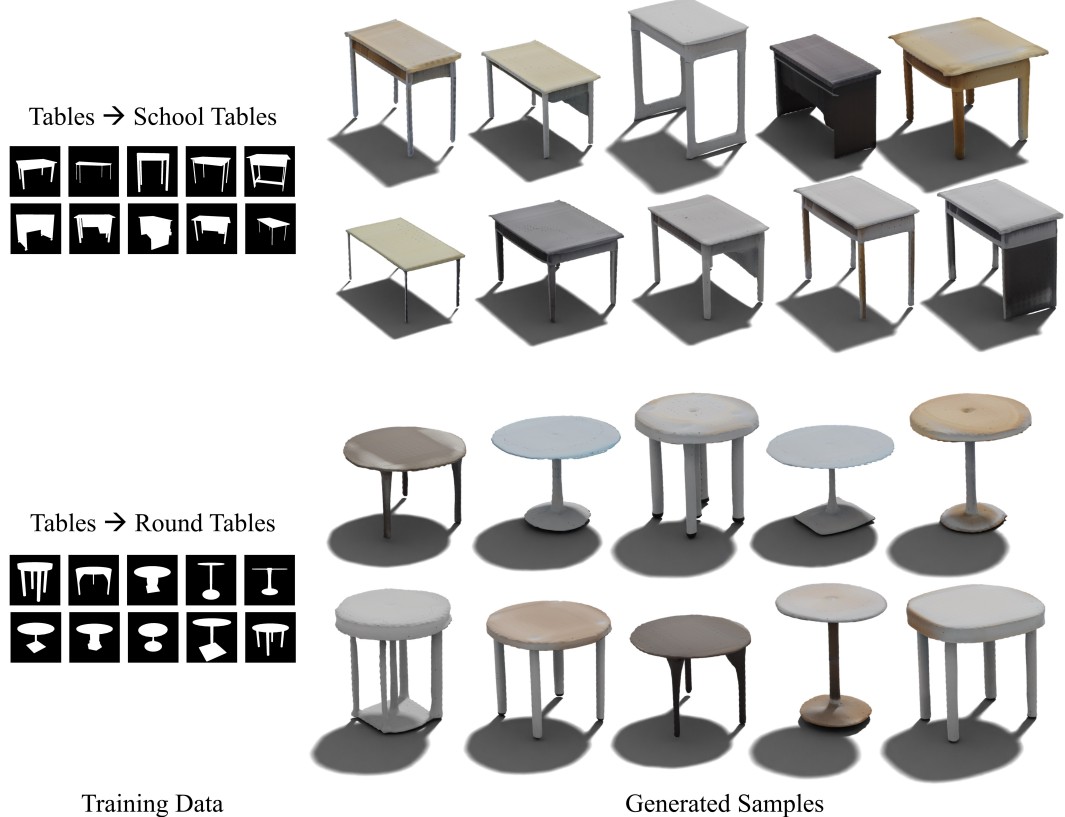

Tables → School Tables

Tables → Round Tables

Training Data                                          Generated Samples

Figure 18: 10-shot generated shapes of our approach using ShapeNetCore Tables as the source domain.

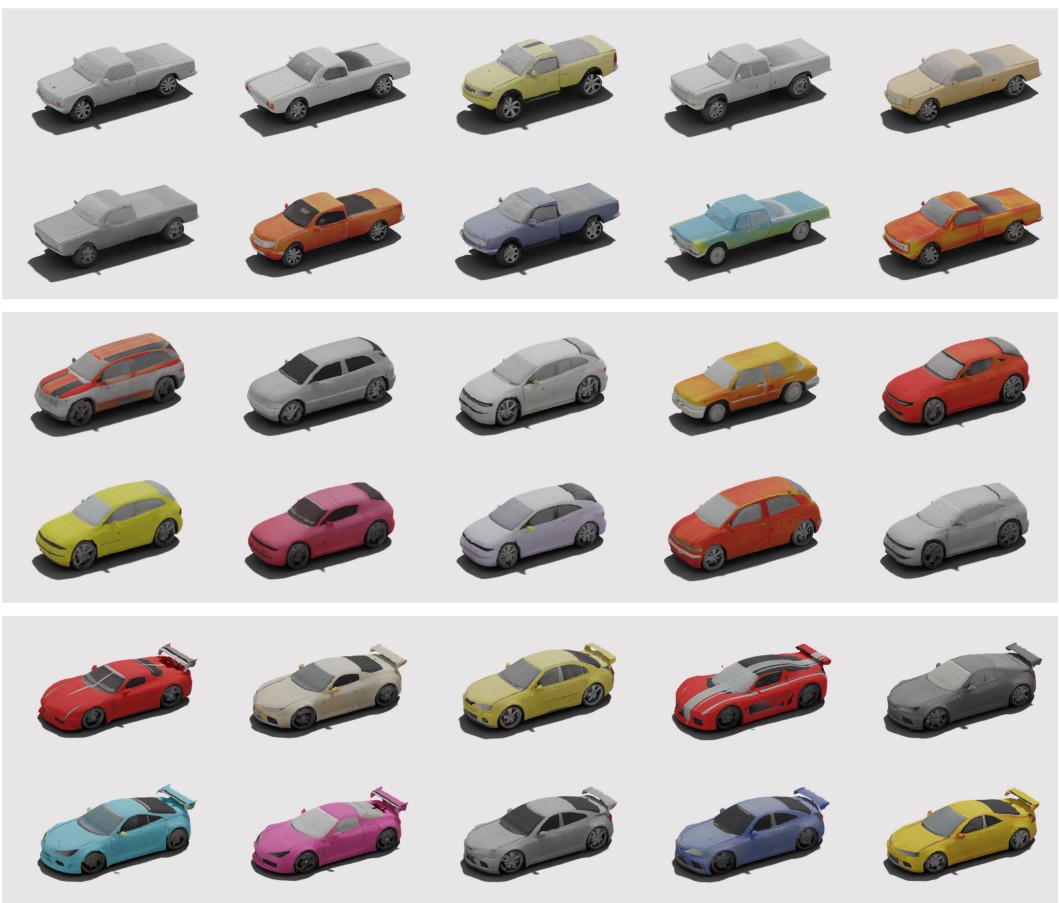

Figure 19: Additional 10-shot generated shapes of our approach on Cars → Trucks, SUVs, and Racing Cars.

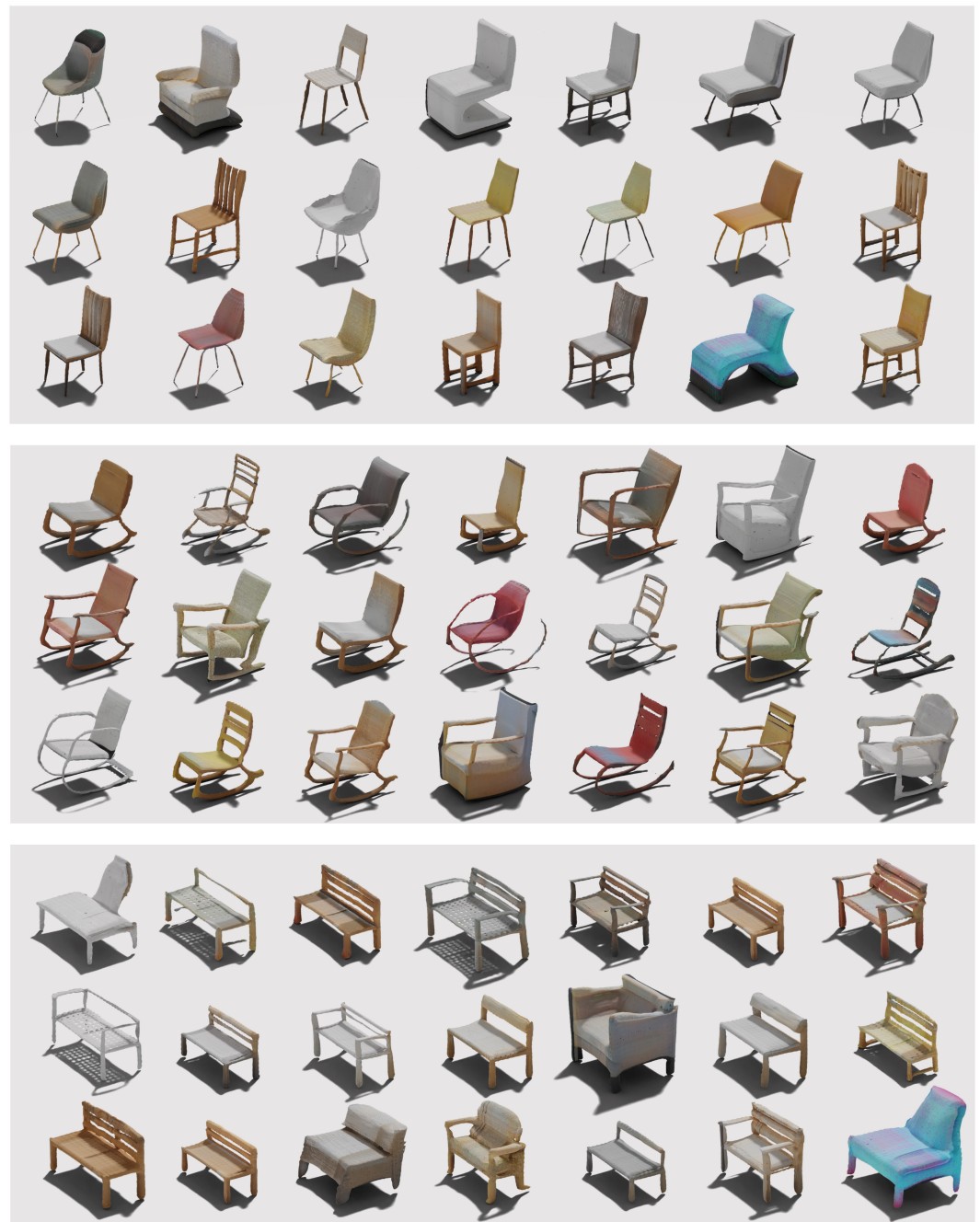

Figure 20: Additional 10-shot generated shapes of our approach on Chairs → Modern Chairs, Rocking Chairs, and Lawn Chairs.

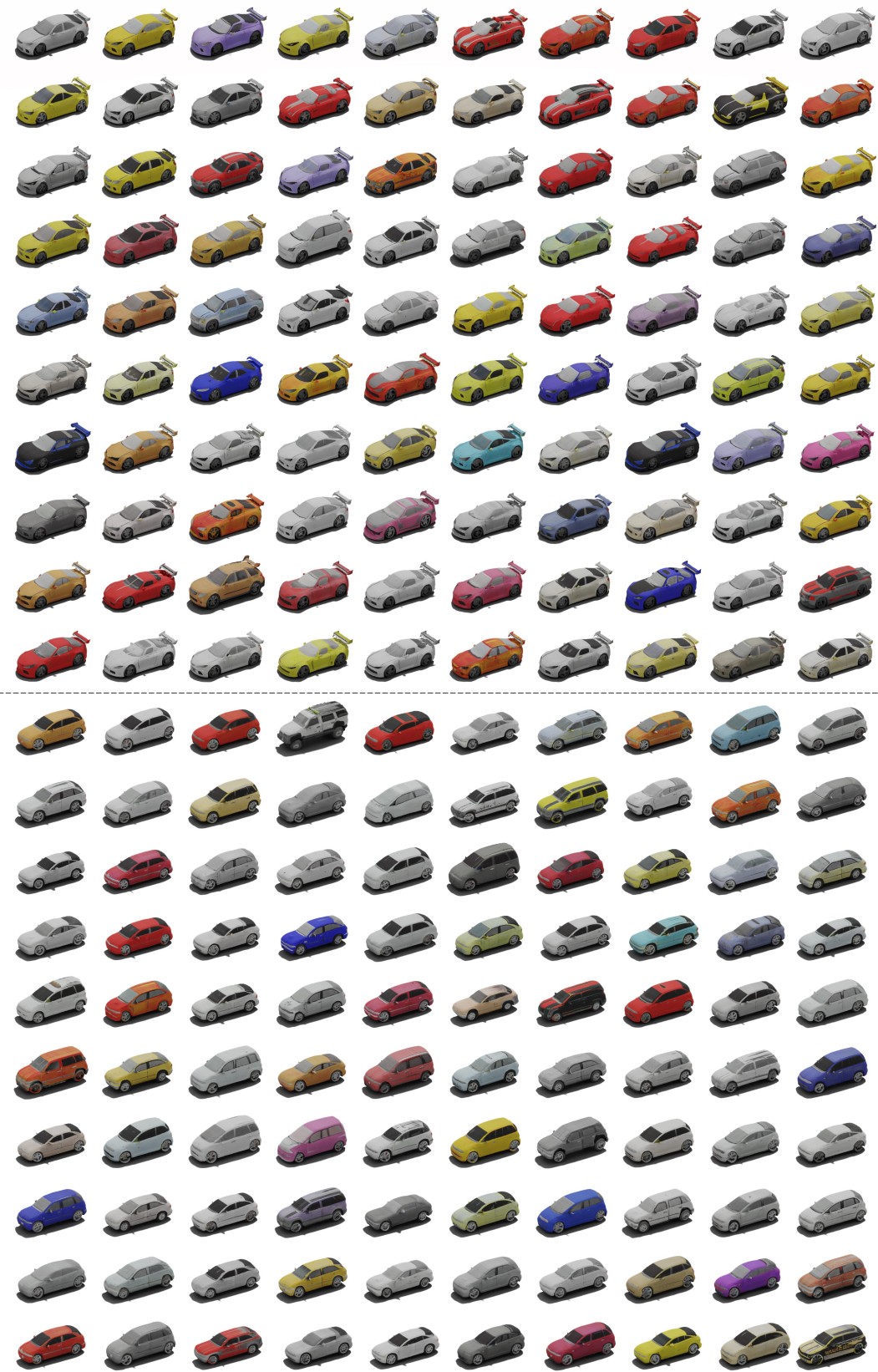

Figure 21: 200 randomly 10-shot generated samples produced by our approach on Cars → Racing Cars and SUVs.

