# OpenReview forum: "Cross-domain Adaptation for Few-shot 3D Shape Generation"
_ICLR.cc/2024/Conference — ICLR 2024 Conference Desk Rejected Submission_

### Official Review · Reviewer_DPcy · 2023-10-30

**Soundness:** 2 fair
**Presentation:** 3 good
**Contribution:** 3 good
**Rating:** 6
**Confidence:** 3

**Summary:**

This work addresses the problem of few shot domain adaptation for 3D shape generation in which a pre-trained source model is adapted to perform 3D shape generation on a target domain given only a few target training samples. A pairwise relative distance regularization is proposed at both the feature and shape level to help prevent the adapted model from overfitting to the few training samples. Additionally, 2D silhouettes are only needed instead of ground truth 3D shapes for learning the geometry distribution of the target domain.

**Strengths:**

1) Proposes an approach to the unexplored problem of few shot domain adaptation for 3D shape generation. This is an interesting problem as 3D generative models are particularly difficult to learn from limited training data. Since many categories are wildly disparate in the number of training samples, domain adaptation is a sensible approach to tackling this training sample imbalance.

2) The paper is well written and easy to follow.

3) Reasonable baselines are proposed to compare the method against, and are shown to be outperformed by the proposed method qualitatively and quantitatively. In terms of generation quality and diversity, the qualitative samples seem to show greater diversity in terms of geometry and more realistic texture adaptation in most cases.

**Weaknesses:**

1) $S_{geo}^{s}$, $S_{geo}^{t}$, $S_{tex}^{s}$, $S_{tex}^{t}$ are never defined in any section of the paper, instead they are only defined in Figure 2. These should also be defined in the paper so it is easier to make sense of Eqs. 1-2 & Eqs. 9-10.

2) In the “cars -> trucks” examples in Fig. 3, it looks like the back window of a car gets painted on to the bed of the truck, suggesting a potential limitation of the approach in which it may not be properly learning to adapt textures for all mappings between source and target domains. This can also be seen in Fig. 1, where the proposed approach has a window as a truck bed while the directly fine tuned models have actual truck beds.

3) While the method does show strong results for adapting source models to a target domain, the only two source domains the method is evaluated on is the Car and Chair categories from ShapeNet. It would be nice to see some results on adapting source models from other categories.

4) It would be good to also include the quantitative ablation (Table 4 in supplemental) in the main paper which better demonstrates how the losses affect generation quality and diversity across an entire dataset rather than just the few samples shown in the qualitative ablation in Fig. 7.

**Questions:**

1) Does the batch size of 4 mentioned in the setup correspond to $N$ describing the number of sampled geometry and texture codes? If so, how does the choice of $N$ affect results, as a batch of samples seem to be used in modeling the geometry and texture pairwise similarity distributions of source and target domains at each training iteration. Is 4 samples really enough to model such distributions?

---

> ### Author Response · Authors · 2023-11-11
> **Response from authors**
>
> Thanks for your precious time for reviews. Here we provide corresponding responses and results to cover your concerns.
>
> 1.$\textbf{Notations}$: We have added the definition of the four functions to the revised manuscript near Eq.1-2, and 9-10. Thanks for your advice.
>
> 2.$\textbf{Textures}$: Since we only use silhouettes as training samples, it is difficult for our approach to get perfect textures. It is designed to maintain similar texture distributions of source samples, leading to some unrealistic textures in some cases. We also provide the method of learning both geometry and texture information from few-shot data in Appendix C. As the first work to tackle few-shot 3D shape adaptation, we hope to achieve better texture adaptation in future work. The directly fine-tuned models are fine-tuned on textured shapes. They overfit and replicate training samples compeletely, leading to more "realistic" textures than ours.
> Besides, there are mature methods to generate textures given geometry structures. Recent 3D generation works like MeshDiffusion and DiffusionSDF only synthesize the geometry structures and generate textures with additional methods or models. It is also convenient to fix some problems in textures with modern software for finer results. Moreover, as discussed in Appendix B, GET3D also produces some low-quality features, which may influence the visualized results in this paper.
>
> 3.$\textbf{Additional Experiments}$:
> We have conducted experiments on Tables and added the results to the revised manuscript in Appendix L.
>
> 4.$\textbf{Quantitative Ablations}$:
> We have moved the quantitative ablations of our approach to the main paper in the revised manuscript following your advice. Thanks!
>
> 5.$\textbf{Batch Size}$: Yes, N corresponds to the batch size. The N latent codes are sampled randomly in every iteration. We set N=4 following most prior few-shot adaptation works. We also tried a larger N of 16 and got very similar results to N=4. Adapted models are guided to learn from amounts of batches (about 3K-5K in our experiments) of N randomly sampled latent codes to model its distributions of pairwise similarities. In every iteration, we employ N=4 samples to keep the relative distances between them while guiding these samples to get close to the target samples. The key idea is to keep the generation diversity during domain adaptation. We empirically find that N=4 works well for the adaptation setups in our paper.
>
> Please let us know if our responses solve your concerns. If you still have any unclear parts about our work, please let us know as well. Thanks for your review.

---

> ### Comment · Reviewer_DPcy · 2023-11-21
>
> Thank you for the detailed response. Most of my concerns have been addressed.
>
> One concern I still have is with the added results on large domain gaps. While I expect the results to look worse than adapting between smaller domain gaps, the results shown seem to have significantly less diversity in terms of geometry. In particular, many of the generated samples are highly similar, while only being similar to 1 or 2 of the few-shot training examples. This makes me question whether maintaining pairwise relative distances makes sense when adapting across larger domain gaps, where the level of diversity in these domains may vary significantly. For example, say you have a source domain with little diversity (where objects may be similar at the feature level) and are adapting to a target domain with greater diversity. If you are only learning the target geometry through adversarial learning, maintaining pairwise relative distances seems like it would encourage mode collapse. This sort of seems like what is happening in the results in Figure 14, where the source Table samples are considerably less diverse than the target modern chairs.

---

> ### Author Response · Authors · 2023-11-21
> **Response from authors**
>
> Thanks for your response. Here we add further explanation about our approach on large domain gaps.
> 1. We randomly displayed some samples in the original Fig. 14. Firstly, we add additional examples to Fig. 14 to the revised manuscript to show that our approach achieves similar diversity as the training samples of modern chairs. Furthermore, we add quantitative results of adaptation from different source datasets in Table 7 in the revised manuscript. It can be seen that the adaptation from tables achieves relatively lower diversity than the adaptation from chairs. However, it still achieves apparently greater diversity than directly fine-tuned models on Chairs $\rightarrow$ 10-shot Modern and Lawn Chairs.
> 2. Our approach is designed to adapt source models pre-trained on large source domains to target domains using limited data. It is mainly designed for related source/target domains. We add experiments to show that it is compatible with relatively larger domain gaps like Tables $\rightarrow$ Chairs. Compared with samples adapted from the model pre-trained on Chairs, we get lower quality with some incomplete detailed structures and some blurred textures. For some target shapes very different from source samples, it is harder for the proposed approach to produce similar results (similar to the mode collapse mentioned in your response), resulting in lower diversity. Despite this, it achieves reasonable and diverse target shapes.
> 3. Our approach proposes to keep relative distances to avoid adapted models from overfitting to training samples and replicating them directly, which is not wanted by generative tasks. Adapted models are encouraged to learn the common features of target domains by our approach instead of replicating them directly. For domain adaptation methods, the generation diversity is limited by source models inevitably. However, our approach could still serve as an effective approach to avoid overfitting and maintain diversity when the diversity of source models is relatively limited, as shown by the better quantitative results than directly fine-tuned models on related source/target domains in Table 7.
> 4. As the first work to tackle the few-shot 3D shape adaptation task, we have provided a practicable solution for related source/target domains. It is also compatible with source/target domains of relatively larger gaps. We will work on better methods to achieve better results in domain adaptation with large domain gaps like Tables $\rightarrow$ Chairs or even larger gaps like Cars $\rightarrow$ Planes in future work.
>
> We hope the explanation can solve your concern. Looking forward to your response.
>
> Thanks for your positive feedback and precious time in reviewing our paper.

---

> ### Author Response · Authors · 2023-11-22
>
> Dear reviewer DPcy,
>
> We thank you for your valuable reviews and positive feedback. We have revised the manuscript and provided further response to cover your concern of our approach on large domain gaps. Today is the last day for discussion between authors and reviewers. We are looking forward to your reply.
>
> Thanks for your review!

---

### Official Review · Reviewer_KzHF · 2023-10-30

**Soundness:** 3 good
**Presentation:** 3 good
**Contribution:** 2 fair
**Rating:** 5
**Confidence:** 3

**Summary:**

This paper presents a few-shot 3D shape generation method that adapts a pre-trained 3D generative model to the domain of the provided few examples (e.g, 10 shapes of trucks). Since the data is limited, directly fine-tuning the model on the few examples can lead to severe overfitting. To address the issue, the paper proposes an additional regularization loss that perserves the pairwise distances between the samples at feature-level and shape-level. This additional regularization loss is the main technical contribution of the paper. The authors evaluate the method on ShapeNet cars and chairs categories and show the improvement over baseline methods.

**Strengths:**

- The presentation of the paper is quite clear, making it easy to understand.
- The paper addresses a specific problem and solves it in an simple and effective approach.

**Weaknesses:**

- My biggest concern is that the gap between the source and target domains seems small, making the problem less interesting. For examples, trucks seem to be only slightly off the distribution of common cars. Is the source model not able to generate trucks at all? In particularly, the source datasets should be made more clear. In page 7 experiment setup, are the types of cars in the target datasets (e.g., trucks) also included in the source datasets? If not, then could you show some nearest neighbors in the source datasets? This would help understanding the gap between source and target datasets.
- The paper claims to be the first few-shot 3D shape generation, but there are indeed 3D generation methods pre deep learning era that only need a few input shapes, e.g. [1]. The first "few-shot 3D adaptation" could be more accurate.
- I don't think being able to train only on the silhouettes can be claimed as an advantage (it seems like so in the paper's tone), as it basically ignores the texture and assumes the texture distribution the same as source domain.
- I'm not super familiar with the few-shot domain adaptation literature so I cannot really judge the technical novelty. It seems to be an adaptation of methods from image domain, but I'll seek to other reviewers examination on this.
- One particular technical point that I doubted is Eqn. 10, where the masks of 2D rendered shapes are used for texture features. How could the rendered masks be used for the feature maps with the correct alignment? You can render from any angle and the resulting masks would be different. I don't get the rational for masking features this way.



[1] Fit and diverse: set evolution for inspiring 3D shape galleries, SIGGRAPH 2012.

**Questions:**

See above.

---

> ### Author Response · Authors · 2023-11-11
> **Response from authors**
>
> Thanks for your precious time for reviews. Here we provide corresponding responses and results to cover your concerns.
>
> 1. $\textbf{Domain Gaps and Datasets}$: The source model is trained on Cars in ShapeNetCore v1 dataset. There are only 31 truck samples in all 3533 car samples, making it hard to generate truck samples with the source model. Besides, the GET3D model is unconditional and cannot be controlled. We cannot say that the source model cannot produce trucks at all. However, we aim to guide target models to generate target samples only. As shown in the visualized samples, the adapted models synthesize trucks while the source model gets other cars with the same noise inputs. We also add CLIP results of the source samples and adapted samples in Appendix K to prove this illustration quantitatively.
> We have provided some detailed explanations in Appendix G. Actually, it doesn't influence our approach's evaluation even if the few-shot target shapes are directly picked from the source datasets. The source models still fail to produce diverse target samples, which is the target of the adapted models obtained through our approach. Instead, the source models generate lots of samples out of target domains. Prior 2D GAN-based few-shot image generation works directly use samples in FFHQ to adapt the source model pre-trained on FFHQ to a smaller target domain like babies or people wearing sunglasses.
>
> 2. $\textbf{More Accurate Description}$: Thanks for your advice, we have changed the "few-shot 3D shape generation" related to "first" to "few-shot 3D shape adaptation" in the revised manuscript to ensure its accuracy.
>
> 3. $\textbf{Learning Geometry Only}$: Our approach is mainly designed for geometry learning, which is mainly considered in 3D shape generation. Our approach is designed to further democratize 3D shape generation. With our approach, only few-shot silhouettes without textures are needed to realize 3D shape generation. We also provide the method of learning both geometry and texture information from few-shot data in Appendix C. As the first work to tackle few-shot 3D shape adaptation, we hope to achieve better texture adaptation in future work.
> Besides, there are mature methods to generate textures given geometry structures. Recent 3D generation works like MeshDiffusion and DiffusionSDF only synthesize the geometry structures and generate textures with additional methods or models. It is also convenient to fix some problems in textures with modern software for finer results. Moreover, as discussed in Appendix B, GET3D also produces some low-quality features, which may influence the visualized results in this paper.
>
> 4. $\textbf{Novelty Discussion}$: We have discussed the novelty of our approach in Appendix F. The proposed losses share similar formats with CDC Ojha et al. (2021) and prior contrastive learning methods but still have significant differences. Firstly, CDC only uses features in generators for the computation of losses. Our approach uses both feature-level and shape-level (image-level) information and achieves higher-quality and more stable results compared with using feature-level losses only, as shown in the ablation results. Secondly, our approach is designed for 3D shape generators and applies to geometry and texture information separately. Thirdly, we propose the concept of shared masks to alleviate the influence of geometry for texture adaptation, which helps our approach better preserve the texture information of source samples during domain adaptation. Besides, the novelty of this work also comes from the research topic. This is the first work to study the few-shot 3D shape adaptation task. In addition, we also introduce several metrics for few-shot 3D shape generation evaluation.
>
> 5. $\textbf{Explanation of Feature Maps}$: The texture generator in GET3D produces tri-plane feature maps. Any point in the generated shape can be projected into the tri-plane feature maps to synthesize its textures. Given a view to synthesize masks, we can extract the corresponding features in the tri-plane feature maps, which are used to synthesize RGB images (including textures) in correspondence to masks with the same camera parameters (angles) in GET3D. That's the reason why we can apply masks to texture features in Eq. 10. GET3D uses RGB images and corresponding masks for 2 different discriminators respectively.
>
> Please let us know if our responses solve your concerns. If you still have any unclear parts about our work, please let us know as well. Thanks for your review.

---

> > ### Comment · Reviewer_KzHF · 2023-11-19
> > **Reply**
> >
> > Thanks for effort and detailed response from the authors.
> >
> > 1. It's hard to check what has been updated in the paper. It would be better if the authors could highlight the revised part of the paper with a different color.
> > 2. Regarding domain gaps, it is good that authors bring up literature in 2D image domain as an example but I'm still not fully convinced.
> >     - So the key problem is there are not sufficient truck examples in the dataset. Will the problem be solved if we have a larger dataset, e.g., objaverse?
> >     - From the response, it seems like the task is mainly about adapt the source model to a smaller domain that is within the domain of original training data. Then it more feels like personalized generative model, e.g., DreamBooth[1], which definitely should be discussed in the paper. Applying DreamBooth on a large 3D diffusion model, e.g., 3DGen[4], could just accomplish the same task.
> >     - I have this concern mainly becuase there are several previous works on shape translation (e.g., table -> chair), such as LOGAN[2] and UNIST[3]. Their domain gap is much larger than what the authors presented here, though they are not few-shot methods. These works should also be discussed.
> >
> > [1] Ruiz, Nataniel, et al. "Dreambooth: Fine tuning text-to-image diffusion models for subject-driven generation." CVPR, 2023.
> > [2] Yin, Kangxue, et al. "LOGAN: Unpaired shape transform in latent overcomplete space." ToG, 2019.
> > [3] Chen, Qimin, et al. "UNIST: unpaired neural implicit shape translation network." CVPR, 2022.
> > [4] Gupta, Anchit, et al. "3dgen: Triplane latent diffusion for textured mesh generation." ArXiv, 2023.

---

> ### Author Response · Authors · 2023-11-20
> **Response from authors**
>
> Thanks for your positive response. Here we add further explanation and experiments for your concerns.
>
> 1.$\textbf{Key Idea}$: The key idea of our work is to preserve the diversity of source models while adapting to target domains to get diverse results using limited data. With a larger dataset, the source models would have greater diversity, resulting in greater diversity in the few-shot adaptation model produced with our approach. For example, if a source model produces 100 different car examples, our approach transfers it to 100 truck examples. If a source model produces 1000 car examples, our approach transfers it to 1000 truck examples. Our approach is designed to adapt pre-trained source models to a related target domain for diverse results with few-shot data (e.g., 10-shot silhouettes). If we have a large dataset of target samples (e.g., 1000 truck samples), we can directly train a new generative model on this dataset to produce diverse truck samples. However, this work focuses on few-shot tasks, and it would be an efficient method to obtain diverse target 3D shapes using limited data. Users only need to provide 10-shot silhouettes for domain adaptation instead of collecting large amounts of 3D shapes as data.
>
> 2.$\textbf{DreamBooth}$: DreamBooth shares different targets with this work. DreamBooth is subject-driven and aims to synthesize novel scenes of the same subject shown in few-shot training samples, for example, novel views or novel contexts. The input includes the pre-trained Stable Diffusion Model and a reference set of the target subject. The output is a fine-tuned Stable Diffusion Model, which could synthesize novel scenes of the target subject following corresponding text prompts. The key features of target subjects are preserved. For example, given a reference set of a dog, DreamBooth synthesizes images of the same dog.
>
> Our work follows prior few-shot image generation methods and is domain-driven. It is designed to generate high-quality and diverse samples of target domains. We aim to extract the common features of limited data and maintain generation diversity by adapting source samples to target domains. The subjects in adapted samples are a lot more diverse than few-shot training data. The input includes the source model pre-trained on a large source domain and few-shot samples of target domains. The output is a fine-tuned model which could generate target samples. For example, we use silhouettes of 10-shot SUVs as training data and adapt the source model of cars to the target domain of SUVs. The adapted model is not trained to synthesize SUVs in training data. Instead, it is trained to generate diverse SUVs (cars sharing common features with few-shot training data).
>
> Besides, 2D DreamBooth is not qualified for domain-driven tasks. For example, it cannot learn the domain knowledge (e.g., styles) from few-shot data. $\textbf{We have added visualized samples of DreamBooth on domain-driven tasks and this discussion to Appendix F in the revised paper.}$ It's hard for DreamBooth to learn domain knowledge and link it to an identifier like [V]. Therefore, applying DreamBooth to 3DGen may fail to accomplish this task.
>
> 3.$\textbf{LOGAN and UNIST}$ are based on VAEs trained on enough data from two domains (e.g., table and chairs). Then translators are trained to transfer samples from one domain to the other based on the latent space provided by VAEs. They tackle a different task from this work and aim to build a translation between two domains. Our approach aims to produce diverse results given few-shot data, the domain adaptation is our method instead of our target. Besides, LOGAN and UNIST are not qualified for few-shot data since both VAEs and translators need enough data for training to avoid overfitting. $\textbf{This discussion has been added to the related work part}$. Thanks for your advice!
>
> 4.$\textbf{Our approach on large domain gaps:}$ Based on your review, we further add experiments of our approach on Tables $\rightarrow$ Modern Chairs and Lawn Chairs (10-shot silhouettes as training data). The visualized results have been added to Appendix E. $\textbf{It can be seen that our approach is also qualified for domain gaps like Tables $\rightarrow$ Chairs used in LOGAN and UNIST.}$ We get diverse chair samples using the model pre-trained on Tables and fine-tune it on 10-shot silhouettes of chairs. In addition, we add quantitative results of the adaptation setups of larger domain gaps in Appendix E as well.
>
> 5.$\textbf{We have highlighted the modifications in blue in the revised manuscript.}$
>
> Looking forward to your respond. Thanks for your review again!

---

> ### Author Response · Authors · 2023-11-22
>
> Dear reviewer KzHF,
>
> We thank you for your valuable reviews and positive feedback. We have revised the manuscript and provided further response to cover your concerns. Today is the last day for discussion between authors and reviewers. We are looking forward to your reply.
>
> Thanks for your review!

---

### Official Review · Reviewer_4te8 · 2023-10-31

**Soundness:** 2 fair
**Presentation:** 3 good
**Contribution:** 2 fair
**Rating:** 3
**Confidence:** 5

**Summary:**

This paper proposes a recently develop 3D model generator known as Get3D to synthesize images of 3D objects that will be helpful in domain adaptation. The synthesized images are evaluated using standard metrics. The problem is also set in a few short setting. Fidelity to shape and texture is emphasized.

**Strengths:**

Good looking pictures
Few shot setting
Comparison with two other methods.

**Weaknesses:**

Experiments are not complete. Since domain adaptation is the motivation, the effectiveness of synthesized data for domain adaptation using Office-Data (Saenko, et al ECCV 2010), or DomainNet data should be evaluated. The loss functions used for texture should be compared with Style-GAN and diffusion-based approaches. It is not clear how effective this approach will be for synthesizing objects at different poses.

**Questions:**

Conduct experiments on Office Data to validate the usefulness of synthetic data for domain adaptation task.

---

> ### Author Response · Authors · 2023-11-11
> **Response from authors**
>
> Thanks for your precious time for reviews. However, we find that you have some misunderstanding of our work. Here we provide corresponding responses and results to cover your concerns.
>
> 1. $\textbf{Task:}$  Our work is the first few-shot 3D shape adaptation work. It is a 3D shape generative task using limited data to synthesize textured 3D shapes instead of synthesizing images of 3D shapes or synthesizing images for domain adaptation tasks. We need to point out that the "cross-domain adaptation" is not the motivation. Instead, it is the method to achieve our motivation of few-shot 3D shape generation. We propose to adapt generative models pre-trained on source domains to target domains using limited data, pursuing high-quality and diverse synthesized textured 3D shapes. The cross-domain adaptation method in our work is different from the traditional domain adaptation task for object recognition, which is usually evaluated with 2D image datasets containing many categories of objects like Office Data or DomainNet.
>
> 2. $\textbf{Extensive Experiments:}$ We agree with you that the experiments for evaluating the effectiveness of the synthesized 3D shapes are necessary. CD, FID and Intra-LPIPS results are provided in our paper to evaluate generation quality and diversity. Here we add more. Instead of evaluating our data with the traditional domain adaptation models trained on Office data (about 4k images) or DomainNet (about 600k images), we add experiments to evaluate the synthesized data with the CLIP model trained on 0.4 billion text-image pairs. We randomly sample 1024 3D shapes and get 24 views of each as the synthesized datasets. We compute the cosine similarity between the embeddings encoded from these images by the CLIP image encoder and the embeddings encoded by the CLIP text encoder with text (e.g., “a photo of a truck”). The results of training datasets and source samples are provided as reference.
>
> $$\\begin{array}{c|c|c|c} \\hline
> \\text{Target Domains} & \\text{Source Samples} & \\text{Adapted Samples (ours)} & \\text{Few-shot Data (Reference)}
> \\\\ \\hline
> \\text{Trucks} & 0.2041 & \pmb {0.2477} & 0.2568 \\\\\
> \\text{SUVs} & 0.2183 & \pmb{0.2548} & 0.2686 \\\\
> \\text{Rocking Chairs} & 0.2549 & \pmb{0.2979} & 0.3120\\\\
> \\text{Lawn Chairs} & 0.2280 & \pmb{0.2742} & 0.2891 \\\\
> \\text{Ambulances} & 0.2175 & \pmb{0.2913} & 0.2961 \\\\
> \\text{Police Cars} & 0.2323 & \pmb{0.2737} & 0.2730 \\\\ \\hline
> \\end{array}$$
>
> It can be seen that our approach significantly improves the consistency between adapted samples and target domains. We achieve similar results to the few-shot training samples, apparently better than source samples. These results have been added to the revised manuscript in Appendix K. Thanks for your advice!
>
> 3. $\textbf{Texture Loss}$: The texture loss is a cross-domain adaptation loss designed to maintain the diversity of textures. It shares different targets with StyleGAN or diffusion losses, which aim to learn from target domains. These methods cannot be compared directly. The employed model GET3D is a 3D shape GAN, we cannot apply the StyleGAN loss or diffusion loss to it as well. We provide detailed ablation analysis to analyze the effect of the proposed losses.
>
> 4. $\textbf{Poses}$: In our experiments, all the training and generated samples (3D shapes) are placed on a platform and sampled with certain camera modes, which are inherited from GET3D. The formats of training samples of our approach should be consistent with the source models. GET3D is a 3D shape GAN designed to synthesize complete 3D shapes instead of different views of 3D shapes like Nerf. Our approach makes it possible to produce diverse results using few-shot silhouettes as data. It is a different task from synthesizing objects with different poses. We don't need to consider poses since we synthesize complete 3D shapes. We can simply get different poses of the synthesized 3D shapes with different camera parameters.
>
> Please let us know if our responses solve your concerns. If you still have any unclear parts about our work, please let us know as well. Thanks for your review.

---

> ### Author Response · Authors · 2023-11-22
>
> Dear reviewer 4te8,
>
> We thank you for your valuable reviews. We have revised the manuscript and provided detailed response to cover your concerns. Today is the last day for discussion between authors and reviewers. We are looking forward to your reply.
>
> Thanks for your review!

---

### Official Review · Reviewer_xeiS · 2023-10-31

**Soundness:** 3 good
**Presentation:** 3 good
**Contribution:** 3 good
**Rating:** 6
**Confidence:** 4

**Summary:**

This paper proposes a method for few-shot 3D shape generation. In particular, they introduce a domain adaptation approach to transfer a pretrained 3D generative model to a similar target domain. The key idea for domain adaptation part is to maintain the relative distances between generated samples at both feature and shape level as similar to that in the source domain. The proposed method only requires silhouettes during training. The method is implemented based on GET3D and show effectivenss on generating new samples with diverse shapes but similar textures with the source domain.

**Strengths:**

- This paper investigates an interesting problem of few-shot 3D shape generation and is the earliest work working on this direction. This may shed new light on transferring pretrained 3D generative models to other domains where very limited data is available.

- The generated shapes look promising in term of shape quality and diversity. While the geometry can achieve variations in the target domain, the texture appearance can maintain similar distribution with the source domain.

- The exposition of the paper is clear and informativie.

**Weaknesses:**

- Despite the good results in the paper, the domain gap between the source and target domain is relatively small. The current domain transfer is limited to the same category, e.g. within cars or within chairs. This limits the application of the proposed approach.

- Though the proposed method claims that it can be applied to other network architectures, I am not fully convinced. The current network design is highly coupled with that of GET3D. To support the claim, additional experiments of using other generative models are required.

**Questions:**

- For Equation (4) and (5), it is very reasonable to use cosine similarity and softmax function to compute the relative distance between the binary masks. Since masks are binary, there may exist simpler but more effective methods to compute the relative distances. For example, have you tried to compute based on the features from the common regions of the two masks?

---

> ### Author Response · Authors · 2023-11-11
> **Response from Authors**
>
> Thanks for your precious time for reviews. Here we provide corresponding responses and results to cover your concerns.
>
> 1. $\textbf{Related domains}$: As the first few-shot 3D shape adaptation work, our approach is designed for related source/target domains. As illustrated in Sec. 5, we also view it as a limitation and hope to achieve domain adaptation for large domain gaps in future work. We have added experimets on relatively large domain gaps like Tables $\rightarrow$ Chairs in Appendix E and show that our approach is qualified for such domain gaps to get diverse target samples using 10-shot silhouettes as training data.
>
> 2. $\textbf{Application to Future Models}$: This work designs loss functions for 3D shape GANs without modifying the network structures to realize few-shot generation. We only need to extract features and 2D images (masks) for loss computation, which makes it possible to apply our approach to other 3D shape GANs in the future. The rendered 2D images and the features can be extracted from any 3D generative models. We only implement our approach with GET3D since it is the only 3D shape GAN that synthesizes textured 3D shapes directly from random noise inputs yet.
>
> 3. $\textbf{Equations 4 and 5}$: Eq. 4 & 5 are designed to keep the relative distances between two masks at image-level. Eq. 1 & 2 are designed to keep the relative distances between the features in the geometry synthesis networks (mask is one of the representations of geometry shapes). The geometry features correspond to the whole 3D geometry structure synthesis, making it impossible to identify which parts of features correspond to the common regions of two masks.
>
> Please let us know if our responses solve your concerns. If you still have any unclear parts about our work, please let us know as well. Thanks for your review.

---

> ### Author Response · Authors · 2023-11-22
>
> Dear reviewer xeiS,
>
> We thank you for your valuable reviews. We have revised the manuscript and provided detailed response to cover your concerns. Today is the last day for discussion between authors and reviewers. We are looking forward to your reply.
>
> Thanks for your review!

---

### Author Response · Authors · 2023-11-11
**Reponse from authors**

We thank all reviewers for your valuable advice. We have provided responses to all concerns proposed by reviewers and provided a revised manuscript of our work following suggestions from reviewers KzHF and DPcy. The main paper and the appendix are combined together.

Besides, we find that several reviewers are confused about our task setting. Here, we provide a more detailed explanation. This paper is the first few-shot 3D shape adaptation work. It is designed to synthesize high-quality and diverse target samples. In the main paper, we propose our approach to learning geometry only from few-shot silhouettes of training samples. The distributions of textures are maintained from source samples, which may lead to unrealistic results in some cases. Our approach democratizes 3D shape generation with few-shot and simple inputs. We also provide another approach to learn both geometry and textures from textured training samples in Appendix C.

We hope that our responses can solve your concerns. If you have any unclear parts about our work or any suggestions on our paper, please let us know as soon as possible. Thank all reviewers for your precious time.

---

### Author Response · Authors · 2023-11-16

Dear reviewers,

We have provided the rebuttal for your reviews. Could you please look at our responses and let us know what concerns are still not solved or if you still have any unclear parts about our work? We are looking forward to more discussion to improve this paper.

Thanks for your precious time!

---

> ### Author Response · Authors · 2023-11-21
> **About Two Days Left For Discussion**
>
> Dear all reviewers,
>
> There are about two days left for discussion. We have provided the rebuttal for your reviews. Could you please look at our responses and let us know what concerns are still not solved or if you still have any unclear parts about our work? We are looking forward to your responses.
>
> Thanks for your review!

---

> > ### Author Response · Authors · 2023-11-22
> > **One Day Left for Discussion**
> >
> > Dear all reviewers,
> >
> > There is only one day left for discussion. Could you please look at our responses and let us know what concerns are still not solved or if you still have any unclear parts about our work? We are looking forward to your responses sincerely.
> >
> > Thanks for your review!

---

> ### Author Response · Authors · 2023-11-23
>
> Dear all reviewers,
>
> There are only about 4 hours left for discussion. Could you please look at our responses and let us know what concerns are still not solved or if you still have any unclear parts about our work? We are looking forward to your responses sincerely.
>
> Thanks for your review!